

# Revised and updated geospatial monitoring of twenty-first century forest carbon fluxes

David A. Gibbs[1], Melissa Rose[1], Giacomo Grassi[2], Joana Melo[2], Simone Rossi[2,3], Viola Heinrich[4], Nancy L. Harris[1]

[1] World Resources Institute, Washington, DC, 20002, USA
[2] European Commission, Joint Research Centre (JRC), Ispra, Italy
[3] Arcadia SIT, Vigevano, Italy
[4] Helmholtz GFZ German Research Centre for Geosciences, Potsdam, Germany & School of Geographical Sciences, University of Bristol, Bristol, UK.

*Correspondence to*: David A. Gibbs (david.gibbs@wri.org)

**Short Summary**

Updated global maps of greenhouse gas emissions and sequestration by forests from 2001 onwards using satellite-derived data show that forests are strong net carbon sinks, capturing about as much $CO_2$ each year on average as the United States emits from fossil fuels. After reclassifying fluxes to countries' reporting categories for national greenhouse gas inventories, we found that roughly two-thirds of the total net flux from forests is anthropogenic and one-third is non-anthropogenic.

**Abstract**

Forests are a key component of climate change mitigation strategies because they both emit and remove atmospheric carbon dioxide. Earth observation data are increasingly used to estimate the magnitude and geographic distribution of greenhouse gas (GHG) fluxes and reduce overall uncertainty in the global carbon budget, including for forests. Here we report on a revised and updated geospatial, Earth observation-based forest carbon flux modelling framework that maps GHG emissions (Gibbs et al. 2024a), carbon removals (Gibbs et al. 2024b), and the net balance between them (Gibbs et al. 2024c) globally from 2001 onwards at roughly 30-meter resolution (Harris et al. 2021, hereafter referred to as the Global Forest Watch (GFW) model). Beyond updating the model to include the most recent years, revisions address some of the original model's limitations, improve model inputs, and refine the uncertainty analysis. We found that between 2001 and 2023, global forest ecosystems were, on average, a net carbon sink of -5.5 ± 8.1 (one standard deviation) gigatonnes $CO_2$ equivalent $yr^{-1}$ (Gt $CO_2$e $yr^{-1}$), which reflects the balance of 9.0 ± 2.7 Gt $CO_2$e $yr^{-1}$ of GHG emissions and -14.5 ± 7.7 of carbon removals, with an additional -0.20 Gt $CO_2$e $yr^{-1}$ transferred into harvested wood products. Uncertainty in gross removals was greatly reduced compared to the original model due to refinement of temperate secondary forest carbon removal factor uncertainties.



To increase the conceptual similarity between fluxes from the GFW model and national greenhouse gas inventories (NGHGIs)
and further policy relevance, we translated (re-allocated) GFW's estimates of gross emissions and removals into fluxes from
forest land and deforestation, i.e. the land use categories that countries use to report anthropogenic forest-related fluxes from
managed land in their NGHGIs. We estimated a global net anthropogenic forest-related sink of -3.5 Gt $CO_2$e yr$^{-1}$ (-3.7 Gt $CO_2$e
yr$^{-1}$ including transfers into harvested wood products). Emissions from deforestation ranged between 3.3 and 5.0 Gt $CO_2$yr$^{-1}$
and the net anthropogenic sink in managed forest land ranged between -6.8 and -8.5 Gt $CO_2$e yr$^{-1}$, reflecting ambiguity about
the reporting category to which countries assign emissions from loss of secondary forests within shifting agriculture systems.
We categorized the remaining net flux of -2.2 Gt $CO_2$e yr$^{-1}$ reported by the GFW model as non-anthropogenic (0.37 Gt $CO_2$e
yr$^{-1}$ emissions and -2.5 Gt $CO_2$e yr$^{-1}$ removals). The magnitude of the GFW model's annual average deforestation emissions
and the global anthropogenic forest sink aligned well with aggregated NGHGIs, although their temporal trends differed;
NGHGIs reported a slightly increasing forest land sink and fluctuating deforestation emissions, while the GFW model reported
a declining sink and increasing deforestation emissions.
Updates to the model and the revised uncertainty analysis demonstrate a spatially explicit forest carbon flux monitoring
framework that is increasingly transparent, operational, timely, and flexible enough to answer research and policy questions.
Moreover, the translation of Earth observation-based flux estimates into the same reporting framework as countries use for
NGHGIs can help build consensus on land use carbon fluxes, support the independent evaluation of progress towards Paris
Agreement goals, and assist national policymakers in locating sources and sinks of forest carbon and their drivers.
**1 Introduction**
Land is among the most uncertain components of the global carbon cycle (Friedlingstein et al. 2023). The highly dynamic and
bi-directional nature of forest carbon fluxes, both spatially and temporally, as well as the contributions of anthropogenic and
non-anthropogenic processes, cause unique challenges for monitoring fluxes. Top-down atmospheric observations, e.g. from
sensors such as NASA's Orbiting Carbon Observatory, are not precise enough to attribute fluxes to specific drivers, and the
current suite of bottom-up approaches for estimating global terrestrial carbon fluxes (Friedlingstein et al. 2023) is based on
models that are not fully consistent with each other (i.e., bookkeeping models and dynamic global vegetation models (DGVMs)
to estimate anthropogenic and natural fluxes, respectively). An additional complication is that these models separate
anthropogenic and natural fluxes from land differently from how national greenhouse gas inventories (NGHGIs) do, which are
used within climate policy treaties and to drive national climate actions. This makes it difficult for models to provide estimates
directly relevant to climate policy frameworks and national climate action. Top-down atmospheric approaches do not make
this separation, while global estimates of anthropogenic land use fluxes from bookkeeping models (Friedlingstein et al. 2023)
are 6.7 Gt $CO_2$ yr$^{-1}$ higher than aggregate NGHGIs (Grassi et al. 2023). This gap is due primarily to definitional and conceptual
differences around what is classified as anthropogenic vs. natural fluxes from forests (Grassi et al. 2018), with recent studies





focusing on reconciling these differences (e.g., Schwingshackl et al. 2022, Grassi et al. 2023). Thus, despite improved data
acquisition and advances in modelling capabilities, large uncertainty and variation in estimates of land emissions and sinks
remain. Moreover, the spatial distribution of forest emissions and, even more so, forest carbon removals are not well
understood, impeding the ability of a range of actors, such as governments, companies, and civil society, to monitor the
effectiveness of land-based climate mitigation actions that reduce emissions from forest loss and maintain or increase forest
carbon sinks.
To address some of these limitations, Global Forest Watch (GFW) introduced an Earth observation-based framework and
model for estimating forest carbon fluxes globally (Harris et al. 2021) that aligns with calls for geospatial monitoring of forest
carbon fluxes (EC 2018; Nyawira et al. 2024; Ochiai et al. 2023; Turubanova et al. 2023). It was designed to fill a gap among
existing forest carbon monitoring approaches by combining global forest change maps, benchmark carbon density maps, and
other Earth observation data based on the IPCC Guidelines for National Greenhouse Gas Inventories (IPCC 2006, IPCC 2019)
that countries use to estimate emissions and removals for their NGHGIs. Within the scope of the Agriculture, Forestry, and
Other Land Uses (AFOLU) sector, only GHG fluxes from forest-related land uses and land-use changes (forest remaining
forest, non-forest converted to forest, forest converted to non-forest) were included. The framework was designed around the
UNFCCC guiding principles for NGHGI preparation: transparency, accuracy, completeness, comparability and consistency.
All GFW carbon flux model inputs and outputs and code are publicly available.
Recognizing that both Earth observation and ground data increase and improve through time, we designed GFW's flux
framework and the model implementing it with the flexibility to accommodate updates to existing components and add new
components. Here we document updates to the model, report the results from the current version, present a revised uncertainty
analysis, and introduce a new translation of GFW model emissions and removals into NGHGI reporting categories of
deforestation and forest land that provides an Earth observation perspective on forest fluxes conceptually similar to what
countries are expected to report under IPCC guidelines.

## 2 Methods

Harris et al. 2021 includes a detailed explanation of the GFW forest flux monitoring framework, but some key elements are
described here. The framework encompasses gross $CO_2$ emissions from loss of carbon in aboveground and belowground
biomass pools, dead wood, litter, and soil organic carbon in mineral soils due to stand-replacing disturbances, carbon loss from
drainage of organic soils, and methane ($CH_4$) and nitrous oxide ($N_2O$) emissions from forest fires and drainage of organic soils.
Carbon removals include sequestration into aboveground and belowground forest biomass. All model inputs are resampled to
the spatial resolution of a Landsat pixel (0.00025x0.00025°, roughly 30x30 m at the equator), and outputs are generated at the
same resolution. The model uses Landsat resolution because it is the highest resolution for which the global forest change



maps and an aboveground biomass map for the year 2000 are publicly available. Higher resolution maps of forest change and
biomass exist but are not publicly available, only for more recent years, and/or include only certain regions (e.g., Vancutsem
et al. 2019, Yang and Zeng 2023).
The IPCC GHG inventory guidelines, the methodological basis of GFW's forest carbon flux monitoring framework, lay out
two methods by which terrestrial carbon stock changes associated with land use, land-use change, and forestry (LULUCF, part
of the broader AFOLU sector) can be calculated: gain-loss and stock-difference (IPCC 2006). Methods can be applied
according to different Tiers (from 1 to 3) with increasing complexity and presumed accuracy. In the gain-loss method, at a
high level, carbon emissions and removals are calculated separately by multiplying activity data such as forest area lost, gained,
or maintained (ha) by emission or removal factors (t C ha$^{-1}$); the net carbon stock change, or flux, is the difference between
gross emissions and gross removals. In the stock-difference method, carbon stocks are measured during repeated inventories
and the difference between remeasurements is the estimate of net carbon stock change, or flux. GFW's framework employs
the gain-loss approach, in which the activity data and other contextual information are estimated using global, Earth
observation-based maps trained on local ground plot data and/or airborne and spaceborne lidar observations.
GFW's gain-loss modeling approach is initialized in the year 2000 with global maps of carbon densities in five forest ecosystem
carbon pools (Fig. 1). We define forest as follows: 1) >30% canopy cover in 2000 (Hansen et al. 2013) or subsequent tree
cover gain (Potapov et al. 2022), 2) non-zero aboveground biomass in 2000 (Harris et al. 2021), 3) mangroves in 2000 (Giri
et al. 2011), and 4) exclusion of oil palm plantations in 2000 (see Table 2). Within the resulting forest mask, gross emissions
are subsequently mapped based on locations of stand-replacing forest disturbances, while gross removals are mapped based
on locations of forest extent and regrowth. In this system of tracking the forest/non-forest status of individual pixels over time,
the model adheres to IPCC Approach 3 for land representation.
For activity data, rather than combining and reconciling national or regional geospatial forest monitoring data in the limited
places where it exists continuously since 2000, we deliberately use global, independent (non-governmental) data sources to
maintain global consistency and comparability within the framework, recognizing that global data are generally not the most
locally accurate or relevant data, but remain useful for large-scale analyses and potentially for verification purposes of other
approaches. To identify forest loss, the GFW model uses the Global Forest Change (GFC) data of Hansen et al. 2013, updated
annually. Because of the framework's use of GFC, emissions are limited to those from stand-replacing disturbances or other
disturbances severe enough to be detected by GFC. Tree cover gain (Potapov et al. 2022) is gross gain and is not assigned to
a specific year. In the model, forest pixels can have loss only (assigned to a specific year), neither loss nor gain (i.e., no change),
or both loss and gain (although the sequence is unknown). Non-forest pixels can have either tree cover gain or no gain; in the
latter case they are outside the framework as they are non-forest remaining non-forest.





Emission and removal factors likewise use spatially explicit data as much as possible to capture spatial variation in forest
properties and dynamics and move beyond ecozone-level representation of forests. GFW model emission and removal factors
are generally independent of national data sources, with the exception of some removal factors in temperate forests, which are
derived directly from the Forest Inventory and Analysis (FIA) database maintained by the USDA Forest Service (see Harris et
al. 2021 and Glen et al. 2024 for details). The model uses a combination of IPCC default (Tier 1) and localized (Tier 2)
emission/removal factors, with the goal of using more Tier 2 factors over time, just as countries are encouraged to do in their
NGHGIs. (Note that some Tier 1 removal factors come from national forest inventories, particularly USFS FIA data (IPCC
2019).) For example, removal factors in primary forests use IPCC defaults (IPCC 2019, Tier 1), while pre-disturbance (year
2000) aboveground biomass carbon densities use a global benchmark map of woody biomass developed from field data and
remote sensing (Harris et al. 2021, Tier 2). Removal factors are applied in a hierarchy from six sources: mangrove-specific
rates (IPCC 2014a), Europe-specific rates by forest type (combination of Table 4.11 of the updated IPCC Guidelines, FAO
Planted Forest Assessment and factors published in national forest inventories), planted tree rates from the Spatial Database
of Planted Trees (SDPT) Version 2.0 (Richter et al. 2024), US-specific rates by region, forest type and age class derived from
the FIA database (Glen et al. 2024), young secondary forest rates (Cook-Patton et al. 2020), and IPCC default rates for all
other areas (e.g., primary forest, older secondary forest in the tropics and in temperate forests outside Europe and the US)
(IPCC 2019). The framework supports the addition of other geospatial removal factors as they become available. Gross
removals are added to pre-disturbance biomass until the year of loss to determine the biomass in the year of loss. Emission
factors are estimated using a map of tree cover loss drivers (Curtis et al. 2018) and burned area (Tyukavina et al. 2022); the
combination of these determine the extent to which carbon pools (including soil organic carbon in mineral soils) are emitted
by forest disturbance. Emission factors are estimated using "committed" emissions (Hansis et al. 2015) or instantaneous
oxidation (IPCC 2019), whereby carbon loss from all relevant pools is assumed to occur in the year of disturbance rather than
modeling delayed carbon fluxes through time.



**Figure 1. Updated conceptual framework for modeling forest-related GHG fluxes.** The model estimates gross forest-related emissions and removals as the product of activity data and emission/removal factors for each ~30-m pixel. The net forest GHG flux is the sum of gross emissions (+) and removals (-). Text and arrows in orange are portions of the removals methodology that are passed into the emissions methodology.

## 2.1 Changes to GFW model input data

Since the original release of GFW's carbon model framework in 2021, which estimated forest carbon flux results through 2019, we have made several changes to the model inputs because new data were published or existing data were improved (Table 1). These changes keep the model aligned with advances in global Earth observation and address some limitations in the original version but do not change the underlying conceptual framework. The updated geospatial inputs are shown in the



context of all inputs in Table 2. We summarize changes to the input data with respect to extension of the model from 2019 to
2023 (Sect. 2.1.1), changes to activity data (Sect. 2.1.2), and changes to emission and removal factors (Sect. 2.1.3).

**Table 1. Changes to GFW model inputs since the original version (Harris et al. 2021).**

| Framework component (article section) | Original version | Current version | Affects emissions | Affects removals |
|---|---|---|---|---|
| Temporal coverage of tree cover loss (2.1.1) | Tree cover loss through 2019 (Hansen et al. 2013, updated annually on GFW) | Tree cover loss through 2023 (Hansen et al. 2013, updated annually on GFW) | Yes | Yes |
| Temporal coverage of drivers of tree cover loss (2.1.1) | Dominant driver of tree cover loss through 2015 (Curtis et al. 2018) | Dominant driver of tree cover loss through 2023 (Curtis et al. 2018, updated annually on GFW) | Yes | No |
| Temporal coverage of burned area (2.1.1) | Burned area through 2019 | Burned area through 2023 | Yes | No |
| Transfers to harvested wood products (country-level only) (2.1.1) | Transfers to HWP through 2015 (FAOSTAT 2021) | Transfers to HWP through 2021 (FAOSTAT 2024) | No | Yes |
| Temporal coverage of tree cover gain (2.1.2) | 2000–2012 (Hansen et al. 2013) | 2000–2020 (Potapov et al. 2022) | Yes | Yes |
| Burned area extent (2.1.2) | MODIS burned area (Giglio et al. 2018, updated annually) | Tree cover loss from fires (Tyukavina et al. 2022, updated annually) | Yes | No |
| Organic soils extent (2.1.2) | • Indonesia and Malaysia (Miettinen et al. 2016) • Below 40° N (Gumbricht et al. 2017) • Above 40° N (Hengl et al. 2017) | • Indonesia and Malaysia (Miettinen et al. 2016) • Central Africa (Crezee et al. 2022) • Lowland Amazonian Peru (Hastie et al. 2022) • Below 40° N (Gumbricht et al. 2017) • Above 40° N (Xu et al. 2018) | Yes | No |
| Planted tree extent (2.1.2) | Spatial Database of Planted Trees v1.0 (Harris et al. 2019) | Spatial Database of Planted Trees v2.0 (Richter et al. 2024) | Yes | Yes |
| Belowground biomass (R:S ratio) (2.1.3) | Global ratio of 0.26 for belowground carbon to aboveground carbon for non-mangrove forests (Mokany et al. 2006) | Map of ratio of belowground carbon to aboveground carbon for non-mangrove forests (Huang et al. 2021)[1] | Yes | Yes |
| Planted tree removal factors and their uncertainties (2.1.3) | Spatial Database of Planted Trees v1.0 (Harris et al. 2019) | Spatial Database of Planted Trees v2.0 (Richter et al. 2024) | Yes | Yes |
| Older secondary (>20 year) | 2019 Refinement to the 2006 IPCC Guidelines for National | 4th Corrigenda to the 2019 Refinement to the 2006 IPCC | Yes | Yes |





| temperate forest removal factors and their uncertainties (2.1.3) | Greenhouse Gas Inventories, Volume 4, Chapter 4, pages 4.34–4.38 Table 4.9 (IPCC 2019) | Guidelines for National Greenhouse Gas Inventories, Volume 4, Chapter 4, pages 4.18–21, Table 4.9 (IPCC 2023)[2] | | |
|---|---|---|---|---|
| Global Warming Potential (GWP) values (2.1.3) | IPCC Fifth Assessment Report, Table 8.7 (100-year, no climate-carbon feedback) (IPCC 2014b) | IPCC Sixth Assessment Report, Table 7.15 (100-year, no climate-carbon feedback) (IPCC 2022) | Yes | No |

[1] The R:S map was extended outwards to fill gaps in the original map.
[2] Removal factors for other climate domains and ages were not updated.
**Table 2. Geospatial data components and sources currently used in the GFW model.** Updated components and sources are denoted
with an * and *italics*. This updates Table S3 in Harris et al. 2021.

| Model component | Source |
|---|---|
| **Forest extent 2000** | |
| Tree cover extent | Hansen et al. 2013 |
| Mangrove forest extent | Giri et al. 2018 |
| Tropical humid primary forest extent | Turubanova et al. 2018 |
| Intact forest landscapes (boreal/temperate) | Potapov et al. 2017 |
| *Planted tree extent (plantations and tree crops)* | *Richter et al. 2024 (Spatial Database of Planted Trees v2.0)* |
| *Peatland extent* | Miettinen et al. 2016 (Indonesia and Malaysia) <br> *Crezee et al. 2022 (Congo Basin)* <br> *Hastie et al. 2022 (Amazonian Peru)* |
| | Gumbricht et al. 2017 (<40° N) |
| | *Xu et al. 2018 (≥40° N)* |
| Oil palm extent 2000 | Austin et al. 2017 (Indonesia) |
| (areas excluded from model) | Gaveau et al. 2014 (Borneo) |
| | Miettinen et al. 2016 (Sumatra, Borneo) |
| | Gunarso et al. 2013 (peninsular Malaysia) |
| **Carbon density 2000** | |
| Aboveground live woody biomass density | Updated from Zarin et al. 2016 (non-mangrove) |
| | Simard et al. 2019 (mangrove) |
| *Belowground biomass density ratio* | *Huang et al. 2021 (root:shoot ratio for non-mangrove forests), with Mokany et al. 2006 filling in gaps* |
| Soil organic carbon density | Hengl et al. 2017 (non-mangrove) |
| | Sanderman et al. 2018 (mangrove) |
| Ecological zone (for deadwood and litter) | FAO 2012 |
| Elevation (for deadwood and litter) | Farr et al. 2007 |
| Mean annual precipitation (for deadwood and litter) | Fick and Hijmans 2017 |
| **Activity data** | |
| *Tree cover loss* | *Hansen et al. 2013 (2001–2023)* |
| *Tree cover gain* | *Potapov et al. 2022 (2000–2020)* |
| *Burned areas* | *Tyukavina et al. 2022 (tree cover loss from fires, updated through the year 2023)* |





| Emission factors | |
|---|---|
| *Drivers of forest loss* | *Curtis et al. 2018 (updated through year 2023)* |
| Climate zone | FAO 2012 |
| Fire combustion and emission factors | IPCC 2019 (Tables 2.5 and 2.6) |
| **Removal Factors** | |
| Ecological zone | FAO 2012 |
| Mangrove removal factors | IPCC 2014a (Wetlands Supplement, Tables 4.4 and 4.5) |
| US forest type | Ruefenacht et al. 2008 |
| US stand age | Pan et al. 2011 |
| US removal factors (by region x type x age class) | Forest Inventory and Analysis Program |
| Europe forest type | Brus et al. 2011 |
| Europe removal factors (by forest type) | IPCC 2019 (Table 4.11) |
| | FAO Planted Forest Thematic Study |
| | Portugal's National GHG inventory |
| *Planted tree removal factors* | *Richter et al. 2024 (Spatial Database of Planted Trees v2.0) (including uncertainties)* |
| Agroforestry removal factors | IPCC 2019 (Tables 5.1 and 5.3) |
| Natural regrowth removal factors (<20 yrs) | Cook-Patton et al. 2020 |
| Primary forest removal factors | IPCC 2019 (Table 4.9) |
| *Old secondary forest removal factors (>20 yrs)* | *IPCC 2019 (Table 4.9 for non-temperate forests only)* |
| | *IPCC 2019/IPCC 2023 (Table 4.9 Corrigenda 4 for temperate forests (including uncertainties))* |
| **Harvested wood products (country only)** | |
| *Production, import and export statistics of sawnwood, wood-based panels and paper & paperboard* | *FAOSTAT (2001–2021)* |


### 2.1.1 Annually updated data

We have updated four inputs to the framework annually since the original GFW model was published: tree cover loss, dominant
driver of tree cover loss, burned area, and country-level transfers to harvested wood products (HWP). In the original version,
they extended to 2019, 2015, 2019, and 2015, respectively. The first three inputs now extend through 2023 and we plan to
continue to update them annually, lagging one year behind the calendar year. Country-level HWP transfers now extend through
year 2021 based on data from FAOSTAT that currently extend through year 2022 (Access date: 5 May 2024). These constitute
the core updates to the model each year.

### 2.1.2 Updated activity data

Beyond the annual updates described above, we have made four additional updates to the model's activity data:



1. Temporal coverage of tree cover gain: Tree cover gain originally covered 2000–2012 but now covers 2000–2020. In the original version, tree cover gain covered seven fewer years than tree cover loss did (12 years of tree cover gain vs. 19 years of tree cover loss); currently, tree cover gain covers three fewer years than tree cover loss (20 years vs. 23 years). Tree cover gain is still reported in one interval, so the framework does not assign gain to a specific year within 2000–2020. The shorter duration of tree cover gain and its lack of information on timing is an ongoing limitation of the inputs to the framework (see Sects. 4.3 and 4.4).

2. Burned area extent: The original version of the GFW model used MODIS burned area (500-m resolution) (Giglio et al. 2018), but now it uses Global Land Analysis & Discovery Lab tree cover loss due to fires (TCLF) (30-m resolution) (Tyukavina et al. 2022). This burned area product is designed to be used with GFC. As in the original version of the model, emissions from fires are included only where stand-replacing disturbances are detected by GFC, meaning that emissions from relatively low severity forest fires remain unquantified in the model.

3. Organic soils extent: We added two new regional tropical peatland maps (Peru and Congo basin, Hastie et al. 2022 and Crezee et al. 2022) and replaced the peat map above 40° N (Xu et al. 2018). These maps reflect a more recent understanding of the extent of organic soils in those regions. This is one of the few inputs to the model that composites regional maps with pan-tropical and global maps.

4. Planted tree extent: Planted trees are part of managed ecosystems, and using distinct removal factors for planted trees instead of removal factors for natural forests better represents the associated carbon sequestration of these managed landscapes. The original version of the GFW model used SDPT v1.0 (Harris et al. 2019) but now it uses SDPT v2.0 (Richter et al. 2024), which includes planted tree extent in 45 additional countries. Richter et al. defines planted trees as plantation forests and tree crops. This dataset aggregates maps of tree crops and planted forests globally in a bottom-up approach that captures roughly 90% of planted tree area globally circa 2020. Each polygon in the database has the most taxonomically resolved information available, from broad type of production (e.g. orchard) to species.

### 2.1.3 Updated emission and removal factors

We have made four updates to emission and removal factors:

1. Belowground biomass (R:S ratio): The original version of the GFW model used a single R:S ratio of 0.26 to estimate belowground biomass applied globally to non-mangrove forests (Mokany et al. 2006) (mangroves had separate ratios from IPCC 2014a). The updated model uses a global R:S map from Huang et al. 2021 to incorporate spatial variability in R:S, ranging from less than 0.15 to greater than 0.5. Because the R:S map does not cover all land where forest is present in our framework (e.g., some near-shore islands), we interpolated missing R:S pixels from nearby ones; where interpolation was not possible (e.g., remote Pacific islands), we retained the original default ratio of 0.26. We applied this ratio map to aboveground biomass in the year of tree cover loss to calculate carbon emissions from loss of belowground biomass. We also used the R:S map to calculate carbon removals by belowground biomass based on





carbon removals by aboveground biomass. Including this input makes the belowground carbon stocks and removal
factors reflect local forest types better than using a single, global ratio.
2.   Planted tree removal factors and their uncertainties: SDPT v2.0 (Richter et al. 2024) has a removal factor and
uncertainty associated with every planted tree (planted forest and tree crop) polygon included in the database. The
removal factors of polygons that were in SDPT v1.0 are largely unchanged in SDPT v2.0, but polygons newly
included in SDPT v2.0 have been assigned removal factors based on information about what kind of planted tree is
present using the most taxonomically resolved information available.
3.   Older secondary (>20 year) temperate forest removal factors and their uncertainties: The original version of the
framework applied Tier 1 removal factors published in Table 4.9 of IPCC 2019 for primary and some secondary (>20
years) temperate forests. In 2023, IPCC released corrected default removal factors and their uncertainties for
temperate secondary forests in North and South America, which are also applied in the GFW model to >20 year old
forests in temperate ecozones outside of the United States and Europe where no better sources of data are currently
available. In the model update, we replaced the original IPCC defaults with the corrected ones.
4.   Global Warming Potential (GWP) values: The original version of the framework converted non-$CO_2$ emissions from
$CH_4$ and $N_2O$ into equivalent units of $CO_2$ using GWP values published in IPCC's Fifth Assessment Report. The
framework now uses GWP values for $CH_4$ and $N_2O$ from IPCC's Sixth Assessment Report. This affects gross
emissions and net flux outputs only where non-$CO_2$ emissions are estimated (organic soil drainage, fires in organic
soils, or biomass burning).

**2.2 Updated uncertainty analysis**

With the original version of the framework, we presented an uncertainty analysis that used an error propagation approach for
inputs for which uncertainties (variances) were available and potentially substantial. This approach underlies Approach 1
(simple error propagation) outlined in the IPCC Guidelines and produces similar results but reflects exact calculations of
variances and standard deviations, whereas IPCC Approach 1 to uncertainty analysis is an approximated approach that yields
95% confidence intervals (IPCC 2019). For the model update, we repeated this uncertainty analysis with all the changes and
updates to the framework described in Sect. 2.1, using the same error propagation approach and the same components as used
in the original analysis.

**2.3 Anthropogenic fluxes from "managed" forests**

GFW's Earth observation-based modelling framework does not (and cannot) differentiate anthropogenic and non-
anthropogenic fluxes from forests. Rather, it includes fluxes from all forest land and therefore the combination of direct
anthropogenic, indirect anthropogenic, and natural fluxes. Thus, results from our model are not directly comparable with those
from NGHGIs or bookkeeping models, each of which define anthropogenic fluxes with different system boundaries for their



specific purposes (Grassi et al. 2022, Grassi et al. 2023). Under UNFCCC decisions and IPCC methodological guidance,
countries report only anthropogenic fluxes in their NGHGIs, approximated by "managed land" (Ogle et al. 2018). Therefore,
if GFW's forest carbon monitoring framework is to serve as an independent, Earth observation-based point of reference for
NGHGIs, its results must be reported in a conceptually similar way covering the same scope. In doing so, we adopted the
proposal of Grassi et al. (2023) in adjusting global data to the NGHGI framework for analyses focused on country policy or
action. In translating the GFW model's fluxes into the NGHGI reporting framework, we did what IPCC guidelines direct
countries to do when compiling and reporting their inventories rather than what countries necessarily do in practice for their
inventories. The goal of this translation exercise was not to reproduce how countries prepare their NGHGIs as closely as
possible using the GFW model, to achieve maximum quantitative similarity to NGHGIs, or to reconcile the GFW flux model
with NGHGIs but rather to present GHG fluxes from a globally consistent, geospatial approach in the same conceptual terms
that national policymakers use.
We developed a three-step process to translate the GFW model's gross emissions and removals into three IPCC reporting
categories: anthropogenic flux from managed forest land, emissions from deforestation (anthropogenic), and non-
anthropogenic flux from unmanaged forest (Table 3). It builds upon the simpler comparison between the GFW model and
NGHGIs conducted in the IPCC Sixth Assessment Report (Nabuurs et al. 2022), in which anthropogenic fluxes from the
former were those outside primary forests in the tropics and intact forest landscapes in the non-tropics. This translation process
does not change the GFW model's bottom-line net flux estimates; rather, it reclassifies the gross fluxes by intersecting the
GFW model fluxes with other contextual geospatial data to provide fluxes more conceptually aligned with those of NGHGIs.
The first step (Sect. 2.3.1) assigned each country to one of three cases based on how their NGHGI applies the managed land
proxy (Fig. 2). The second and third steps reclassified the GFW model's emissions (Sect. 2.3.2) and removals (Sect. 2.3.3),
respectively, into three IPCC reporting categories according to the three cases assigned in step 1 (Fig. 2). Emissions and
removals within each IPCC reporting category were then summed to calculate net anthropogenic and non-anthropogenic forest-
related fluxes for each country. The GFW model calculates annual emissions, corresponding to the year of tree cover loss, but
does not calculate annual removals and instead calculates removals as an annualized average over the entire model period.
Thus, to generate timeseries from the GFW model using the NGHGI reporting categories, we calculated the average annual
removals in each reporting category by dividing gross removals by the number of model years. The resulting time series for
each reporting category is therefore the difference between the annual emissions for that year and the average annual removals.
For this analysis, we used data from the GFW model for 2001–2022 to align with the temporal coverage of NGHGIs. Because
the GFW model cannot currently report emissions from organic soil separately from all other emissions, we combined
NGHGIs' deforestation and organic soil emissions (including emissions from forest land, from peat decomposition and peat
fires typically associated to deforestation, and from agriculture soils) to achieve the same scope as the model. We excluded
transfers into the harvested wood products pool from both data sources in this translation analysis because that is not a core
element of our geospatial framework.





**Table 3. Translating GFW flux model gross emissions and removals to national greenhouse gas inventory (NGHGI) reporting**
**categories.** To calculate total net flux for IPCC reporting categories, GFW flux model emissions and removals were reclassified according
to managed land status (managed vs. unmanaged) and driver of tree cover loss. Following IPCC guidelines, for Case 2 countries we used
information about the driver of tree cover loss to reassign initially delineated unmanaged forest to managed forest where direct human
activity is observed to result in tree cover loss (i.e. forestry, commodity-driven deforestation (CDD), urbanization, and shifting agriculture).
Thus, all associated fluxes from unmanaged forests reassigned to managed forests are reported in the corresponding anthropogenic IPCC
reporting category (anthropogenic forest land flux and deforestation).

| | | Anthropogenic Forest Land Flux *Forest Remaining Forest and Non-Forest Land Converted to Forest* | | | Deforestation *Forest Converted to Non-Forest Land* | | | Non-Anthropogenic Forest Land Flux *Forest Remaining Forest* | | |
|---|---|---|---|---|---|---|---|---|---|---|
| **Step 1: Managed land delineation (2.3.1)** | Country case | 1 | 2a | 2b | 1 | 2a | 2b | 1 | 2a | 2b |
| | Managed land proxy | All forests managed | Managed land polygons | Non-intact/non-primary forests | All forests managed | Managed land polygons | Non-intact/non-primary forests | All forests managed | Managed land polygons | Non-intact/non-primary forests |
| **Step 2: Reclassify gross emissions (2.3.2)** | Managed land | Forestry ✓* Wildfire ✓ Unknown ✓ Shifting ag +¹ CDD ✗ Urbanization ✗ | | | Forestry ✗ Wildfire ✗ Unknown ✗ Shifting ag +² CDD ✓* Urbanization ✓* | | | *N/A* | | |
| | Unmanaged land | *N/A* | | | *N/A* | | | *N/A* | Forestry ✗ Wildfire ✓ Unknown ✓ Shifting ag ✗ CDD ✗ Urbanization ✗ | |
| **Step 3: Reclassify gross removals (2.3.3)** | Managed land | ✓ | | | ✗ | | | *N/A* | | |
| | Unmanaged land | *N/A* | | | *N/A* | | | *N/A* | ✓ | |

✓ = Included
✗ = Excluded
+ = Included in certain scenario


* Includes emissions from not only the initial delineation of managed forests, but also from tree cover loss in unmanaged forests reassigned to managed forests due to direct human
activity.
¹ To calculate the maximum emissions in anthropogenic forest land, we count emissions from shifting agriculture (shifting ag) in secondary forest toward the anthropogenic forest
land flux and emissions from shifting agriculture in primary forests toward deforestation.
² To calculate the maximum emissions from deforestation, we count all emissions from shifting agriculture in both primary and secondary forest toward deforestation. This also
corresponds to a larger sink in anthropogenic forest land.

**2.3.1 Managed land delineation**
In the first step (Table 3), we assigned countries to one of three cases based on careful review of NGHGIs (Melo et al., in
preparation). These cases describe which land is considered managed and unmanaged according to information that countries
provide in their NGHGIs regarding their use of the managed land proxy (Fig. 2). Case 1 included 46 countries (primarily



UNFCCC Annex 1 countries, i.e. advanced economies with annual GHG reporting commitments) that explicitly consider all
forest land managed and another three countries (China, India, Indonesia) for which we assumed that all forest land is
considered managed, based on the information provided in their NGHGIs. Case 2 included all other countries, which do not
consider all forest to be managed and thus consider some forest to be unmanaged. For the three Case 2a countries (Brazil, the
United States, and Canada), we used the georeferenced boundaries of managed and unmanaged lands that they use in their
NGHGIs. The remaining 143 countries (UNFCCC non-Annex 1 countries, i.e. countries with historically less stringent GHG
reporting commitments) either report no information or not enough details regarding the use of the managed land proxy and
its extent. For example, Russia's inventory explicitly includes unmanaged land but reports areas by administrative unit rather
than spatially, which is not adequate for our analysis. For these Case 2b countries, we approximated managed forest in tropical
regions as forests outside humid tropical primary forests from 2001 (Turubanova et al. 2018) and in extratropical regions as
forests outside intact forest landscapes from 2000 (Potapov et al. 2017). For Case 2 countries, the initial managed forest
delineation was modified in steps 2 and 3 to include unmanaged land reassigned to managed land due to direct anthropogenic



activity. We note that while countries' definitions of forest land differ, we instead used a single, global definition of forest as
defined in Sect. 2, with a tree cover density >30% (Hansen et al. 2013).

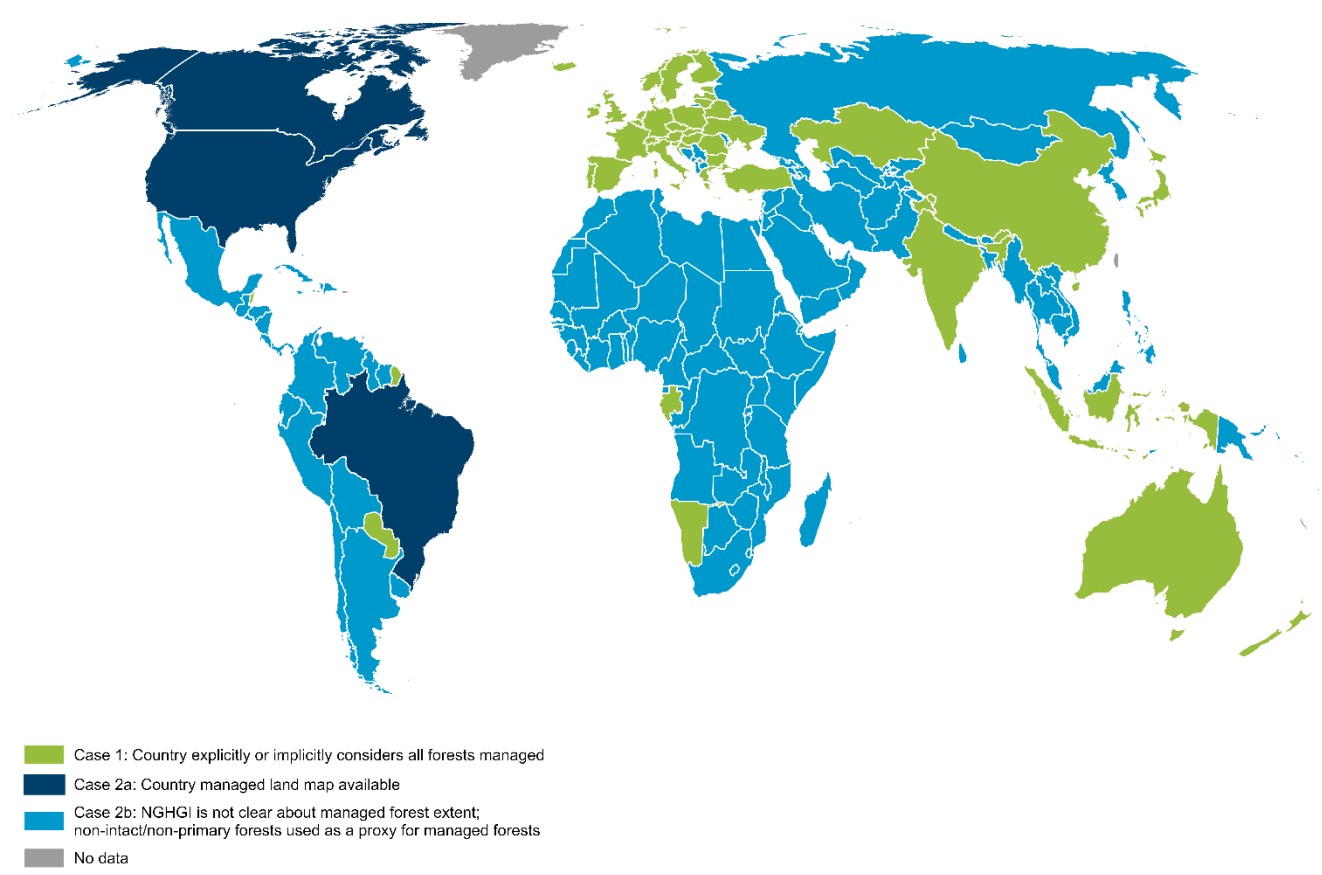


**Figure 2. Country representation of managed land in their national greenhouse gas inventories (NGHGIs).** Countries consider fluxes
by forests in several ways in their national greenhouse gas inventories (Melo et al. in preparation). Some countries explicitly or implicitly
consider all forests to be managed and thus include all forest fluxes in their NGHGIs (Case 1). The rest do not consider all forests to be
managed. Only a few countries (Case 2a) use maps of managed lands to delineate anthropogenic fluxes from non-anthropogenic fluxes. The
rest are not clear in their NGHGIs about the spatial extent to which forests are or are not considered managed and thus which forest fluxes
are included in their inventories (Case 2b).




### 2.3.2 Reclassifying gross emissions


In the second step (Table 3), we combined the initial delineation of managed forests described in Sect. 2.3.1 with a map of
drivers of tree cover loss (Curtis et al. 2018, updated through 2023) to partition the GFW model's gross emissions into IPCC
reporting categories because not all of the GFW model's gross emissions are from deforestation. For Case 1 countries, which
classify all forests as managed, all emissions occurring within country borders were anthropogenic and no emissions were non-
anthropogenic. For Case 2 countries, all emissions within managed forest boundaries (defined in Sect. 2.3.1) were
anthropogenic and the remaining emissions within initially delineated unmanaged forest boundaries were either anthropogenic
or non-anthropogenic depending on the driver of the tree cover loss. We expanded our definition of managed forests to include
initial unmanaged forest as defined in Sect. 2.3.1 where there is a direct human activity, such as forest harvest or deforestation
(IPCC 2006). Thus, we considered all emissions from direct human activity to be anthropogenic. The remaining emissions—
from natural or semi-natural drivers of tree cover loss, such as wildfire, occurring within unmanaged forest boundaries—were
the only emissions we considered to be non-anthropogenic.
Using this delineation of anthropogenic vs. non-anthropogenic, we reclassified the GFW model's gross emissions into three
categories that are conceptually aligned with IPCC reporting categories (Table. 3): anthropogenic emissions on managed forest
land ("forest remaining forest" plus "non-forest land converted to forest"), anthropogenic emissions from deforestation ("forest
converted to non-forest land"), and emissions on unmanaged forest land that are non-anthropogenic by definition ("forest
remaining forest").
Anthropogenic emissions from managed forest land. For all countries, this category included emissions from wildfire and the
small amount of emissions not assigned to a driver (Curtis et al. 2018) occurring within managed forest areas. This category
also included emissions from forestry regardless of where they occurred (inside or outside initial delineated managed land
boundaries as defined in Sect. 2.3.1) because harvest activity is a direct human activity and thus any tree cover loss from
forestry activity results in the reclassification of unmanaged forest to managed forest.
Anthropogenic emissions from deforestation. For all countries, this category was the sum of all emissions from tree cover loss
due to commodity-driven deforestation and urbanization, regardless of where they occurred, as well as emissions from the loss
of intact/primary forests in areas of shifting agriculture because this is a permanent change in land use.
Non-anthropogenic emissions from unmanaged forests. For Case 1 countries, we assumed based on their NGHGIs that all
forests are considered managed and thus no emissions are considered non-anthropogenic. The two categories above represent
all emissions from the GFW model. For Case 2 countries, which have some unmanaged forest (as defined in Sect. 2.3.1), non-
anthropogenic emissions were the sum of the remaining emissions outside managed forests: emissions from tree cover loss
due to wildfires and the (small) unassigned drivers class (Curtis et al. 2018). Although some fires in unmanaged land can be



caused by humans, we classified emissions from them as non-anthropogenic to be consistent with IPCC guidelines; separating
emissions from human-caused fires in unmanaged land and reporting them as anthropogenic forest land emissions could be
improved in further iterations of this analysis.
It is often not clear to which land use categories emissions from shifting agriculture cycles are allocated in NGHGIs, because
this distinction is not required by the IPCC Guidelines (IPCC 2019). Following Curtis et al. (2018), shifting agriculture
landscapes are defined as "small- to medium-scale forest and shrubland conversion for agriculture that is later abandoned and
followed by subsequent forest regrowth." To highlight the sensitivity of how emissions from shifting agriculture landscapes
are estimated, we created two scenarios for our emissions reclassification. In one scenario, we calculated the maximum
emissions from deforestation by including all emissions from the loss of both primary and secondary forests within shifting
agriculture landscapes and therefore no emissions from shifting agriculture are considered to be occurring in forest remaining
forest. In the other scenario, we calculated the maximum emissions from managed forest land by including emissions from the
loss of secondary forests in shifting agriculture landscapes in the anthropogenic forest land flux. This transferred a subset of
emissions considered to be deforestation under the alternative scenario to forest land. The remaining emissions from loss of
intact/primary forests due to shifting agriculture were still considered deforestation emissions, as described above. The two
scenarios do not change the total net anthropogenic forest flux (fluxes from forest land plus deforestation) because the same
emissions are assigned to either category. In both scenarios, emissions from the loss of intact/primary forests due to shifting
agriculture were always classified as deforestation because we considered them to arise from a permanent change from forest
to a non-forest land use.

**2.3.3 Reclassifying gross removals**
In the third step (Table 3), we partitioned carbon removals occurring on forest land as either anthropogenic or non-
anthropogenic. No forest carbon removals were included in deforested land; any removals in pixels with tree cover loss were
assigned to either anthropogenic forest land removals or non-anthropogenic forest removals, as described below. Since
NGHGIs do not treat removals uniformly, we used the three managed land proxy cases to align GFW flux model removal
estimates with how countries report removals in their NGHGIs (Fig. 2).
For Case 1 countries, which explicitly or implicitly consider all forest land to be managed, we classified all removals across
the full GFW model extent as anthropogenic forest land. No removals for these countries were considered non-anthropogenic.
For Case 2 countries, we separated removals into anthropogenic and non-anthropogenic categories following the same spatial
proxy used to delineate managed forests (Sect. 2.3.1). In this approach, we classified all removals in managed forest land as
anthropogenic, including unmanaged forest reclassified as managed forest due to tree cover loss from forestry and shifting
agriculture. All removals in unmanaged forest land were classified as non-anthropogenic.



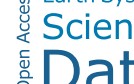

## 3 Results

### 3.1 Emissions, removals, and net fluxes from GFW's updated flux model

In the updated GFW flux model, average annual global gross emissions from stand-replacing forest disturbances were 9.0 Gt $CO_2e$ yr$^{-1}$ between 2001 and 2023 (with 98% from $CO_2$ and 2.4% from $CH_4$ and $N_2O$), average annual gross removals were 14.5 Gt $CO_2$ yr$^{-1}$, and the average annual net forest ecosystem sink was -5.5 Gt $CO_2e$ yr$^{-1}$ (Table 4). Globally, the HWP pool was an additional net carbon sink of -0.20 Gt $CO_2$ yr$^{-1}$, resulting from the transfer of carbon out of forest ecosystems and into the HWP pool. Although the original and revised values in Table 4 are not directly comparable due to different temporal coverage and model updates, it does give a high-level view of the degree to which the collective changes to the model have affected (or not affected) fluxes. Figure 3 maps the updated gross emissions, gross removals, and net GHG flux for forests, and are derived from Gibbs et al. 2024a, Gibbs et al. 2024b, and Gibbs et al. 2024c, respectively.

Our framework allows flexible, yet consistent, estimates of carbon fluxes in a variety of forest types, spatial scales, and regions. Tropical and subtropical forests continued to be the largest contributors to global forest carbon fluxes, contributing 74% of gross emissions (6.7 Gt $CO_2e$ yr$^{-1}$) and 60% of gross removals (-8.8 Gt $CO_2$ yr$^{-1}$). However, temperate forests are the largest net sink, comprising 40% of the global net sink (-2.2 Gt $CO_2e$ yr$^{-1}$). Together, humid tropical primary forests (Turubanova et al. 2018) and intact forest landscapes (Potapov et al. 2017) outside the tropics were a net carbon sink of -0.26 Gt $CO_2e$ yr$^{-1}$ (average annual emissions of 2.8 Gt $CO_2e$ yr$^{-1}$ and removals of 3.1 Gt $CO_2$ yr$^{-1}$). Forests within protected areas (UNEP-WCMC 2024) accounted for 31% (-1.7 Gt CO2e yr-1) of the global net carbon sink. In 2023, gross emissions from Canada's wildfires exceeded emissions from all humid tropical primary forests loss that year (3.0 vs. 2.4 Gt $CO_2e$, respectively; MacCarthy et al. 2024). Updated emissions, removals, and net flux statistics by country and smaller administrative levels can be found on www.globalforestwatch.org.

**Table 4. Forest GHG fluxes by climate domain and globally, with uncertainties expressed as standard deviations, for the original and updated models.** Values in parentheses are the percent of the global flux that occurred in each climate domain. * denotes fluxes with major changes in the uncertainties in the revised GFW model (see Sect. 3.3). The original and updated values are not directly comparable due to different temporal coverage and model updates.

**Forest GHG fluxes Gt CO₂e yr⁻¹ (± standard deviation)**

| Climate domain | Gross emissions | | Gross removals | | Net GHG flux | |
|---|---|---|---|---|---|---|
| | Original *(2001–2019)* | Updated *(2001–2023)* | Original *(2001–2019)* | Updated *(2001–2023)* | Original *(2001–2019)* | Updated *(2001–2023)* |
| Boreal | 0.88 ± 0.42 (11) | 1.4 ± 0.75 (16) | -2.5 ± 0.96 (16) | -2.5 ± 0.95 (17) | -1.6 ± 1.1 (21) | -1.1 ± 1.2 (20) |
| Temperate | 0.87 ± 0.60 (11) | 0.93 ± 0.62 (10) | -4.4 ± 48* (28) | -3.1 ± 0.55* (22) | -3.6 ± 48* (47) | -2.2 ± 0.83* (41) |
| Subtropical | 1.0 ± 0.59 (12) | 1.0 ± 0.93 (11) | -1.6 ± 0.56 (10) | -1.7 ± 0.56 (12) | -0.65±0.81 (8.6) | -0.70± 0.80 (13) |





| Tropical | 5.3 ± 2.4 (66) | 5.7 ± 2.4 (63) | -7.0 ± 7.6 (45) | -7.1 ± 7.6 (49) | -1.7 ± 8.0 (22) | -1.4 ± 7.9 (26) |
| **Global** | **8.1 ± 2.5 (100)** | **9.0 ± 2.7 (100)** | **-16 ± 49* (100)** | **-14.5 ± 7.7* (100)** | **-7.6 ± 49* (100)** | **-5.5 ± 8.1* (100)** |

(a)

Gross forest greenhouse gas emissions
Mt $CO_2$e $yr^{-1}$ (2001–2023)
0.20
0

(b)

Gross forest greenhouse gas removals
Mt $CO_2$e $yr^{-1}$ (2001–2023)
0.0
-0.10

(c)

Net forest greenhouse gas flux
Mt $CO_2$e $yr^{-1}$ (2001–2023)
0.17 (source)
0 (neutral)
-0.83 (sink)

397



**Figure 3. Forest-related GHG fluxes (annual average, 2001–2023).** a) Gross GHG emissions. b) Gross carbon dioxide removals. c) Net GHG flux. Fluxes are aggregated to 0.04 x 0.04° cells for display purposes.

**3.2 Effect of GFW model changes on forest carbon flux estimates**

Updates to the GFW flux model changed gross emissions, gross removals, and net flux over all spatial scales. Average annual gross emissions in the updated GFW model are 12% higher than in the original version, primarily due to higher gross annual emissions since 2019 (8.5 Gt $CO_2$e yr$^{-1}$ between 2001 and 2019 vs. 11.4 Gt $CO_2$e yr$^{-1}$ between 2020 and 2023). Updated gross annual removals are 7.3% lower than in the original model, primarily due to the use of corrected, lower IPCC Tier 1 removal factors for temperate forests, which are applied to 290 Mha of secondary forests in the framework, primarily throughout Eurasia and Canada. Annual average net GHG flux decreased accordingly by 28% from the original version because of both higher emissions and lower removals.

Although we did not quantify the degree to which each change to the model individually affects emissions and removals because we implemented multiple changes simultaneously, we describe how the inputs changed and some general impacts on gross emissions and removals.

*Activity data:*

1. Temporal coverage of tree cover gain: The area of tree cover gain increased globally from 78 Mha in the original version (gain through 2012) to 130 Mha in the current version (gain through 2020). Carbon removals associated with areas of tree cover gain increased from -0.57 to -0.62 Gt $CO_2$ yr$^{-1}$. As in the original model, carbon removals occurring in these young (<20 years) forests remain relatively small compared to gross removals occurring in older, established forests that are much more extensive in total area (96% of gross removals occurred in older forests).

2. Data source for burned area: Use of the new source of fire data with higher spatial resolution (TCLF) combined with an increase in forest fires across Australia, Spain, the United States and Canada between 2020 and 2023 led to an increase of global average annual burned area that coincided with tree cover loss from 4.3 Mha yr$^{-1}$ (2001–2019) to 6.0 Mha yr$^{-1}$ (2001–2023). Global average emissions increased from 1.0 to 1.7 Gt $CO_2$e yr$^{-1}$ in areas where tree cover loss was identified as burned.

3. Data sources for organic soils extent: Improved data led to an increase in the extent of organic soils from 477 Mha to 760 Mha and the area of tree cover loss on organic soils increased from 0.77 Mha yr$^{-1}$ to 2.4 Mha yr$^{-1}$. Emissions from organic soil drainage in areas with tree cover loss increased from 0.21 to 0.91 Gt $CO_2$e yr$^{-1}$, occurring primarily in Indonesia and Malaysia (17% and 3.1% of global total, respectively). Higher emissions from organic soil drainage is due to a combination of increased organic soil extent, planted tree extent, and tree cover loss compared to the original model.

4. Data sources for planted tree extent: Planted forest and tree crop extent increased from 140 Mha to 230 Mha and tree cover loss in planted tree polygons increased from 42 Mha to 64 Mha.



*Emission and removal factors:*

1. Data source for R:S ratios: The previous global R:S used across the full model extent was 0.26. Now, the average ratio of aboveground removals to belowground removals is 0.27 but with considerable geographic variation.

2. Planted tree removal factors and their uncertainties: The average aboveground removal factor in planted trees originally was 3.2 t C ha$^{-1}$ yr$^{-1}$ but using SDPT v2.0 it is 2.3 t C ha$^{-1}$ yr$^{-1}$. Global planted forests and trees were originally estimated to be a net carbon sink of -0.30 Gt $CO_2$e yr$^{-1}$ but using SDPT v2.0 they are now a net sink of -0.54 Gt $CO_2$e yr$^{-1}$, with the increased area of planted trees compensating for the lower average removal factor.

3. Older secondary (>20 year) temperate forest removal factors and their uncertainties: Older secondary temperate forests using IPCC Tier 1 removal factors (i.e., areas affected by this change) originally covered 310 Mha and now cover 290 Mha. Gross removals in these forests declined from -2.7 to -1.3 Gt $CO_2$ yr$^{-1}$.

4. Global Warming Potentials: Updated model results of non-$CO_2$ emissions associated with biomass burning and drainage of organic soils were negligibly impacted by using updated GWPs.

## 3.3 Updated uncertainty analysis

Nearly all changes to the framework are represented in the error propagation approach and therefore affect the global and climate domain uncertainty analyses to some degree. However, the largest change to the uncertainty analysis in terms of input values was the corrected IPCC Tier 1 temperate forest removal factors, which the model applies across large areas of Eurasian and Canadian forests. Some of the largest changes for removal factors and their uncertainties include temperate mountain forest >20 years old (previously 4.4 t aboveground biomass (AGB) ha$^{-1}$ yr$^{-1}$ $\pm$ 100.7 ($\pm$ standard deviation); now 2.1 $\pm$ 0.02 t AGB ha$^{-1}$ yr$^{-1}$) and temperate oceanic forest >20 years old (previously 9.1 t AGB ha$^{-1}$ yr$^{-1}$ $\pm$ 20.2; now 4.9 $\pm$ 0.25 t AGB ha$^{-1}$ yr$^{-1}$). We did not formally assess the contributions of individual model changes to uncertainty because the change in IPCC Tier 1 temperate forest removal factor uncertainties was so dominant.

Uncertainty (reported as one standard deviation) in temperate gross removals declined from 48 Gt $CO_2$ yr$^{-1}$ in the original GFW model to 0.55 Gt $CO_2$ yr$^{-1}$, with uncertainty for gross emissions in this biome increasing slightly from 0.60 to 0.62 Gt $CO_2$e yr$^{-1}$ and uncertainty for net flux decreasing from 48 to 0.83 Gt $CO_2$e yr$^{-1}$ (Table 4). Reduced uncertainty in temperate gross removals propagated to reduced uncertainty in global gross removals and net flux. In the uncertainty analysis for the current version of the model, tropical gross removals has the highest uncertainty, driven by relatively high uncertainty in IPCC's Tier 1 removal factors, which the GFW model applies to tropical primary forests and older secondary forests. Large uncertainties for climate domain and global net flux estimates should be interpreted with caution; their uncertainties are proportionately very large in part because net flux they reflect the sum of negative (removals) and positive (emissions) terms, compounding the addition of their uncertainties.





**3.4 Anthropogenic fluxes from "managed" forests**

When gross emissions and removals from the GFW flux model for 2001–2022 were reclassified into NGHGI reporting categories, the anthropogenic net flux in managed forest land ranged between -6.8 and -8.5 Gt $CO_2e$ yr$^{-1}$ (with and without emissions from shifting agriculture in secondary forests, respectively) and emissions from deforestation ranged between 3.3 and 5.0 Gt $CO_2e$ yr$^{-1}$ (without and with emissions from shifting agriculture in secondary forests, respectively) (Fig. 4, Table A1). The resulting net anthropogenic forest flux—the combined flux from both anthropogenic forest land and deforestation—was -3.5 Gt $CO_2e$ yr$^{-1}$. The non-anthropogenic net sink was -2.2 Gt $CO_2e$ yr$^{-1}$, comprised of -2.5 Gt $CO_2e$ yr$^{-1}$ removals and 0.37 Gt $CO_2e$ yr$^{-1}$ emissions from fires and tree cover loss without an assigned driver in unmanaged forests. The difference in global net flux estimates between the untranslated GFW model (-5.5 Gt $CO_2e$ yr$^{-1}$) and the NGHGI-translated one is that the latter includes only anthropogenic forest-related fluxes in managed land, while the former also includes fluxes from unmanaged land. The combined NGHGI-translated anthropogenic and non-anthropogenic forest flux differs by about 0.2 Gt $CO_2e$ yr$^{-1}$ from the untranslated net flux because the former does not include fluxes from 2023 and does not include fluxes from 32 countries (mostly small island countries), which did not have comparable NGHGIs.

Under the scenario which included emissions from shifting agriculture from secondary forests in deforestation (Fig. 4, hatched bars), GFW's maximum estimate for global deforestation emissions aligned with the combined NGHGI deforestation and organic soil emissions (5.0 Gt $CO_2e$ yr$^{-1}$). In that scenario, GFW's corresponding maximum estimate for global net sink in anthropogenic forest land was larger than estimated by NGHGIs. Under the alternative scenario, which included emissions from shifting agriculture in secondary forests in the anthropogenic forest land flux (Fig. 4, non-hatched bars), GFW's minimum estimate for global net sink in anthropogenic forest land was similar to the NGHGI net forest sink (-6.6 Gt $CO_2$ yr$^{-1}$), but GFW's corresponding minimum estimate for global deforestation emissions was lower than estimated by NGHGIs. The combined GFW flux model net anthropogenic forest sink in managed lands is 1.9 Gt $CO_2e$ yr$^{-1}$ greater than in NGHGIs (-1.5 Gt $CO_2$ yr$^{-1}$).

For Non-Annex 1 countries, the GFW model high and low estimates for forest land and deforestation bracketed the corresponding NGHGI fluxes. However, GFW estimated the net anthropogenic forest flux for Non-Annex 1 countries to be a small net anthropogenic sink while NGHGIs estimates them to be a small net anthropogenic source. For Annex 1 countries, deforestation emissions from the GFW model were much lower than from NGHGIs (0.046–0.049 and 0.55 Gt $CO_2e$ yr$^{-1}$, respectively) and the net forest sink was somewhat larger (-3.1 and -2.3 Gt $CO_2e$ yr$^{-1}$, respectively).

487

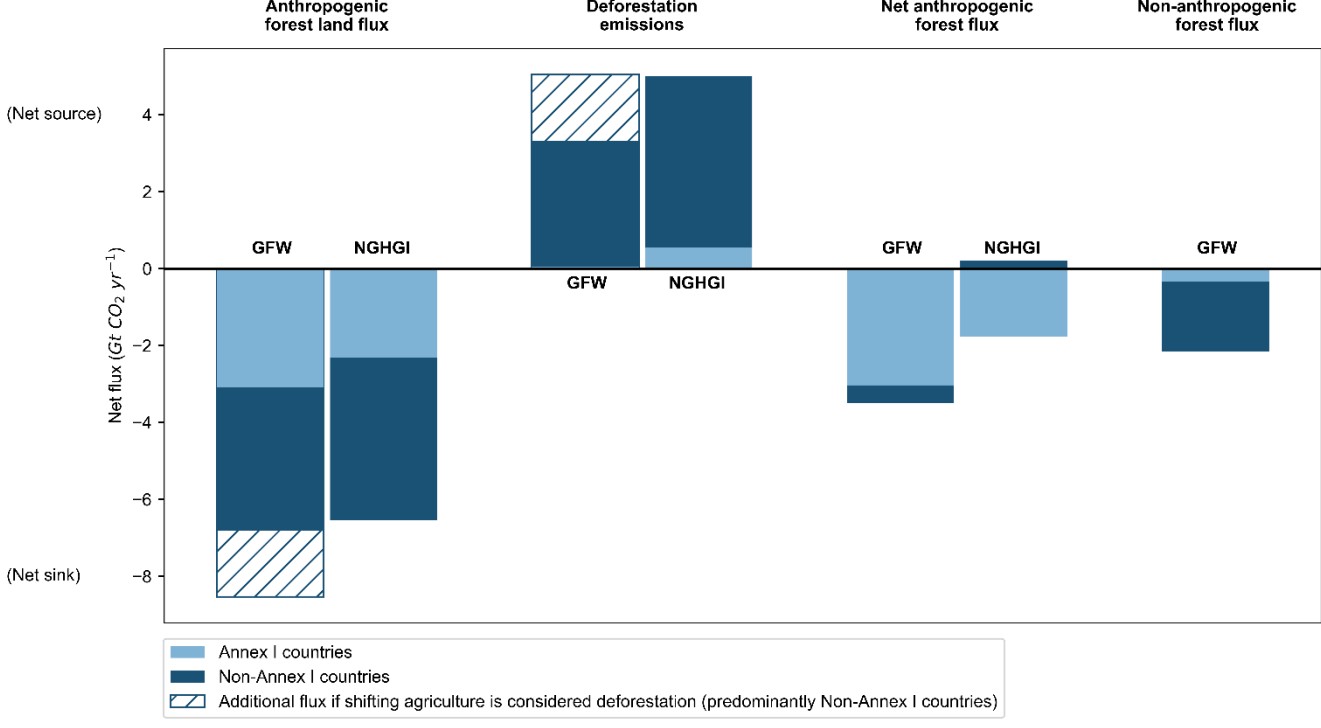

488

**Figure 4. Comparison of average annual forest carbon fluxes (2001–2022) between national greenhouse gas inventories (NGHGI) and the updated GFW flux model.** For the GFW flux model, net anthropogenic forest flux is calculated as the sum of the net anthropogenic forest land flux in managed forests and deforestation (Sect. 2.3). Non-anthropogenic forest flux is calculated as emissions and removals occurring outside managed forests. Because country reporting on emissions from the loss of secondary forests associated with cycles of shifting agriculture is ambiguous, these emissions are shown for the GFW model as hatched bars to indicate how they impact totals depending on the reporting category (forest land or deforestation).

Although the magnitude of the global GFW model estimates for deforestation emissions and the anthropogenic sink in forests align with the aggregated NGHGIs for 2001–2022 under different scenarios, their trends through time do not agree (Fig. 5). Both globally and for Non-Annex 1 countries, the NGHGIs suggest that from 2001 to 2022 forest land became a slightly larger sink and deforestation emissions lacked a clear trend. However, the GFW flux model results suggest the opposite: a reduced sink in forest land and increased deforestation emissions. The forest land flux and deforestation emissions from NGHGIs and the GFW model for Non-Annex 1 countries appear to converge in the last 10 years (roughly -6





Gt CO$_2$ yr$^{-1}$ and 5 Gt CO$_2$ yr$^{-1}$, respectively). For Annex 1 countries, the forest land sink decreased much more according to
the GFW model than NGHGIs, while deforestation emissions stayed fairly constant in both.

**Figure 5. Comparison of forest carbon fluxes timeseries (2001–2022) between national greenhouse gas inventories (NGHGIs) and**
**the updated GFW flux model for Non-Annex 1, Annex 1 countries, and globally.** NGHGI values shown here exclude any fluxes from
harvested wood products, and deforestation emissions are the combined emissions from both deforestation and organic soils to conceptually





align with the scope of fluxes from the GFW framework. For the world and Non-Annex 1 countries, GFW model results are shown in two
timeseries: one where emissions from shifting agriculture in secondary forests is included in that reporting category and one where those
emissions are not included. For the GFW model in Annex 1 countries, the two scenarios are essentially the same and thus we show only one
line.

## 511    4 Discussion

We focus our discussion on the following topics. First, we examine how the updated GFW forest flux model compares with
results from a recent global estimate of forest fluxes by Pan et al. (2024) and the Global Carbon Budget (GCB). Second, we
discuss how fully geospatial, Earth observation-based forest flux estimates can be translated into the reporting categories of
NGHGIs and how transparency in both approaches can result in methodological improvements. Third, we discuss strengths
and limitations of GFW's Earth observation-based forest carbon flux model. Fourth, we outline future research topics which
provide partial solutions to the model's current limitations.

### 518    4.1 Comparison with other recent global flux estimates

Pan et al. (2024) is a relevant comparison for the GFW model because both include only forests and report gross rather than
net fluxes. Pan et al. (2024) estimated gross removals by forests, gross emissions from tropical deforestation, and the global
forest carbon sink by synthesizing forest plot data (inventories and long-term monitoring sites) from 1990 onwards. The
removals estimates are conceptually similar (e.g., both include established and new forests), but the emissions estimates have
different scope (global for GFW, tropical for Pan et al. 2024) (Table 5). The global net fluxes from Pan et al. 2024 and the
updated GFW model are remarkably similar given their entirely different approaches, and thus provide multiple lines of
evidence for a forest sink of around 6 Gt $CO_2$ yr$^{-1}$. Differences in gross emissions and removals between the data sources likely
arise from different scopes and system boundaries, but may be balanced out when combined in the global net flux. Pan et al.
estimated higher tropical gross emissions than the GFW model did for the tropics and subtropics for 2001-2019. When the
GFW model's gross emissions ($CO_2$ only) are limited to the tropics and subtropics and one geospatially implemented definition
of deforestation (tree cover loss due shifting agriculture in primary forest, and all commodity- and urbanization-driven tree
cover loss), it estimates 3.2 Gt $CO_2$ yr$^{-1}$, well below the tropical deforestation estimate of Pan et al. 2024. More broadly
including all tree cover loss in the tropics and subtropics, the GFW model estimates gross emissions of 6.3 Gt $CO_2$ yr$^{-1}$.







**Table 5. Comparison of GFW flux model results to Pan et al. 2024 and the Global Carbon Budget (GCB).** Estimates from the three data sources are not directly comparable due to differences in scope, data, methodologies and reporting structure. GFW model fluxes are limited to 2001–2022 for comparability with the GCB.

| Flux | GFW model, 2001-2022 (Gt $CO_2$ yr$^{-1}$) | Pan et al. 2024, 2000-2019 (Gt $CO_2$ yr$^{-1}$) | Global Carbon Budget, 2001-2022 (Gt $CO_2$ yr$^{-1}$) |
|---|---|---|---|
| Emissions | 8.6 (gross, all observed disturbances)[a] | 7.4 (gross, tropical deforestation)[b] | 4.9 (net, anthropogenic)[c] |
| Removals | -14.7 (gross, all forest ecosystems (-14.5) and HWP (-0.2))[d] | -13 (gross, global) | -11.4 (net, non-anthropogenic) |
| Net | -6.1 (net, all forests)[e] | -5.6 (net, global) | -6.4 (net, all land) |

[a] Gross emissions from all forest disturbances (anthropogenic and non-anthropogenic) observed from Landsat data for the period 2001–2022. Estimate includes $CO_2$ only for comparability with GCB; non-$CO_2$ emissions are 0.19 Gt $CO_2$e yr$^{-1}$. This value is lower than that of Table 4 (9.0 Gt $CO_2$e yr$^{-1}$) because this one includes emissions for 2001–2022 only and excludes non-$CO_2$ gases.

[b] Includes emissions from degradation.

[c] Estimates only net direct anthropogenic effects, including deforestation, afforestation/reforestation and wood harvest. Gross fluxes higher but not reported.

[d] Gross removals from all forest processes (direct, indirect and natural). HWP = transfers to harvested wood products.

[e] Calculated as the net balance between gross forest ecosystem emissions and removals (8.6 – 14.5 Gt $CO_2$ yr$^{-1}$) in this table plus an additional net removal of -0.20 Gt $CO_2$ yr$^{-1}$ into HWP. This value differs from that of Table 4 (-5.5 Gt $CO_2$e yr$^{-1}$) because this one uses lower gross emissions (see note a).

Another point of comparison is the GCB, released by the Global Carbon Project each year. The GCB provides annual estimates of GHG emissions and carbon sinks, when relevant, for all sectors. The GFW flux model is not designed to represent the land portion of the global carbon cycle, nor is it directly comparable with the land use fluxes included in the GCB because of differences in definitions, scope, reporting structure, and methods (Friedlingstein et al. 2023). Three overarching differences are: 1) The GCB reports net sources and sinks for all land (including croplands, grasslands, semi-arid savannas and shrublands), while the GFW model reports gross emissions and removals for forests only; 2) the GCB categorizes fluxes by process into net anthropogenic emissions from land use change and forestry and the "natural" land sink, while the GFW model categorizes fluxes by activity data; 3) the GCB uses global bookkeeping models to estimate anthropogenic carbon fluxes from land use and DGVMs to estimate the natural land sink, while the GFW flux model uses a single integrated approach to estimate emissions and removals. Nevertheless, comparison of the GFW model with the GCB is useful because they use entirely different data sources and approaches, and, as such, convergence between them would represent multiple lines of evidence.

We estimated a global net $CO_2$ sink by forest ecosystems of -6.1 Gt $CO_2$ yr$^{-1}$ between 2001 and 2022, which is similar to the net $CO_2$ land sink of -6.4 Gt $CO_2$ yr$^{-1}$ in the GCB for all terrestrial fluxes over the same period (Table 5). The GCB's net emission estimate (4.9 Gt $CO_2$ yr$^{-1}$) is lower than GFW's gross emissions estimate (8.6 Gt $CO_2$ yr$^{-1}$) partially because the GCB's land-use change emissions (sources) reflect the net balance between anthropogenic emissions and anthropogenic removals associated with forest regrowth. Similarly, the GFW model's gross removals reflect removals across all forest lands,



including removals implicit (but unreported) in the GCB net land-use change estimate (Friedlingstein et al. 2023). Additional
reclassification of fluxes from the GFW model into net anthropogenic from land-use change and the natural land sink may be
possible for further comparisons with the GCB, as has been done between the GCB and NGHGIs (Schwingshackl et al. 2022).
In the comparison of the original GFW model with the GCB, we included a non-spatial estimate of emissions from tropical
forest degradation of 2.1 Gt $CO_2$e yr$^{-1}$ from Pearson et al. 2017 that potentially included some emissions from small-scale
disturbances which we assumed our original model did not capture. For this and subsequent comparisons between the GFW
flux framework and the GCB, we are discontinuing the inclusion of a non-spatial estimate of degradation emissions from a
source external to our framework to maintain its internal consistency and fully geospatial nature. We acknowledge that the
GFW model itself is likely omitting both emissions (e.g., from degradation not detected by TCL) and removals (e.g., from low
canopy density or regenerating forest), but those are gaps that the model should be able to fill over time (see Sect. 4.4). Adding
external data such as Pearson et al. 2017 risks double-counting emissions in the global total. As more geospatial data on
distinguishing deforestation from degradation (Vancutsem et al., 2021) becomes available globally, and geospatial data on the
emission and removal factors associated with forest degradation (Holcomb et al., 2024) and recovery (Heinrich et al., 2023b)
becomes available, it may be possible to reintegrate forest degradation and its associated fluxes.

## 4.2 Translating between Earth observation-based fluxes and NGHGIs

Our goal in translating GFW model results into a NGHGI reporting framework was to provide independent estimates of forest-
based GHG fluxes based on globally consistent, Earth observation-based forest flux data in the reporting categories that
national policymakers use. It was not to reproduce how countries classify their managed land, report their forest fluxes in
practice, or compare fluxes for individual countries. For example, we did not rely solely on the use of managed land polygons
for Case 1 countries to define managed forest; if our observations detected direct human activity in unmanaged polygons, we
assigned those fluxes as anthropogenic forest land fluxes or deforestation. Thus, although this translation makes the GFW
model more conceptually similar with NGHGIs in that the outputs are supposed to represent the same fluxes, they are still not
necessarily entirely comparable because we did not exactly reproduce what countries do in practice within their NGHGIs. It
demonstrates that the GFW model is sufficiently flexible to approximate the system boundaries of anthropogenic fluxes in the
IPCC reporting framework and that Earth observation models can be used to independently monitor anthropogenic GHG fluxes
from forests if adequate country data are made publicly available. The 6.7 Gt $CO_2$ yr$^{-1}$ gap in global land use emissions between
NGHGIs and the GCB has been largely explained (Grassi et al. 2023) and translation between NGHGIs on the one hand and
bookkeeping models and DGVMs on the other is becoming routine (e.g., Schwingshackl et al. 2022); this work is the start of
a similar process for explaining the gap between NGHGIs and Earth observation-based models, primarily through reallocation
of emissions and removals to match NGHGIs' land use categories and filtering the results with maps of managed forest as a
proxy to delineate anthropogenic from non-anthropogenic fluxes.



Although the conceptual alignment produces quantitatively similar annual average fluxes for the GFW model and NGHGIs
globally and for Non-Annex 1 countries, the trends from NGHGIs and the GFW model differ (Fig. 5). For Non-Annex 1
countries, where the trends in each data source are most evident, NGHGIs reported the forest land sink strengthening slightly
while deforestation emissions fluctuated but were generally steady. The GFW model, on the other hand, reported a weakening
sink in forest land and deforestation emissions that increased correspondingly. The tight association between the decreasing
forest land sink and increasing deforestation emissions in the GFW model is due to the use of average annual gross removals
over time (i.e. a constant value), with only gross emissions varying year to year. In NGHGIs, forest land and deforestation can
both change through time and are therefore not driven by the trajectory of just one flux. Quantitative similarity between the
GFW model and NGHGIs may be further improved when the GFW model's gross removals can vary through time as well
(Sect. 4.4). Moreover, for Non-Annex 1 countries, results from the GFW model and NGHGIs have converged for forest land
and deforestation since around 2010, with the two GFW model scenarios bracketing NGHGI fluxes from both reporting
categories after that year. This indicates that the GFW model, and the tree cover loss data that underlies its gross emissions,
were perhaps under-detecting loss as detected by NGHGIs in the early part of the time series.
Exploration of the differences between the GFW model and specific countries' NGHGIs is beyond the scope of this paper;
future work may include more detailed reclassification of the GFW model's fluxes and comparisons with specific regions or
countries. Further sub-setting results from our framework to differentiate anthropogenic and non-anthropogenic fluxes for
comparison with NGHGIs for individual regions, countries and other local-scale analyses is possible and encouraged. Indeed,
comparison of the GFW model and countries' inventories is a way to explore the complementarity and discrepancies between
Earth observation data and inventories, encourage transparency for both, and improve both approaches (Heinrich et al. 2023a).
For example, one advantage of the GFW model, which includes forest fluxes undifferentiated by human contribution, is that
it encompasses both anthropogenic and non-anthropogenic fluxes. When this translation exercise is conducted, GHG fluxes
from managed land can be put in the context of all land fluxes and compared with fluxes from unmanaged land. Because
NGHGIs are not required to estimate fluxes from unmanaged land (just report the area of unmanaged land), aggregation of
NGHGIs does not provide context for managed land fluxes with unmanaged land fluxes. In other words, the GFW model can
indicate the scale of non-anthropogenic fluxes that countries are not reporting in their NGHGIs (which are nevertheless affect
atmospheric $CO_2$ concentrations and global temperature), while NGHGIs are necessary for the GFW model to approximate
the anthropogenic fluxes that are being monitored by countries and the focus of the Paris Agreement. An alternative approach
for reconciling global models and NGHGIs would be for NGHGIs to report all land fluxes in the country, in both managed
and unmanaged land (Nabuurs et al. 2023), but adoption of this seems unlikely.
Future improvements to our flux reclassifications, which may improve regional or country-level comparisons, could include
customizing tree cover density thresholds that align more closely with countries' forest definitions to filter forest extent and
thus the associated fluxes on a country-by-country basis. Additionally, we used maps of primary forests and intact forest





landscapes from 2001 and 2000, respectively, to approximate the extent of unmanaged forests at the initial year of our model
framework. Further refinement to the GFW model's estimates of fluxes from managed lands could include recategorizing
forests as "managed" or "unmanaged" using updated primary/intact forest boundaries in different years to reflect changes to
countries' managed land area over time whenever known. Furthermore, for simplicity, we considered all forest removals as
forest land and did not differentiate the relatively small amount of removals from forest gain as "other land converted to forest",
which is a category that countries report in their NGHGIs. Another improvement would be to separate the emissions from
drainage of organic soils and the emissions from deforestation in the GFW model; in the current translation, deforestation
emissions and organic soil emissions are combined in both data sources. Separating them would further narrow the conceptual
similarity, which would matter most in countries with high emissions from organic soils. Finally, emissions from fires
occurring in unmanaged land could theoretically be differentiated into anthropogenic vs. non-anthropogenic using additional
geospatial data, rather than our simplified assumption that all fires in unmanaged forests are non-anthropogenic in origin.
While our geospatial, Earth observation-based framework permits estimation of fluxes for any geospatially defined forest and
the inclusion (or exclusion) of any area of interest, it cannot distinguish between managed versus unmanaged land without
relevant spatial data. Thus, the ability of the GFW model, and Earth observation models in general, to be translated into IPCC
categories largely depends on the transparency with which countries report on their managed lands. Three countries that
currently apply the managed land proxy (Canada, Brazil, and the United States) have publicly available managed land maps
(Ogle et al. 2018). For all remaining countries, the use and application of the managed land proxy was assumed based on the
available information from country reports. In the absence of this information, primary or intact forest have been used as proxy
for unmanaged forest. With sufficient transparency and flexibility in both the Earth observation-based products and NGHGIs,
the differences between them can be explored.
A crucial driver of forest disturbance, and thus emissions, in the GFW model is shifting agriculture. However, the comparison
between GFW and NGHGI is complicated by the fact that countries typically do not provide specific information on shifting
agriculture in their land representation; according to the IPCC guidelines it can be implicitly included either in forest or in
other land uses (e.g., cropland) (Grassi et al. 2023). Thus, we developed two scenarios for the treatment of fluxes from shifting
agriculture (Fig. 4). Hopefully, as countries begin to submit their Biennial Transparency Reports under the Paris Agreement,
their use of the managed land proxy, the treatment of shifting agriculture, and other exclusions from inventories will be
progressively clarified and translation between approaches will become more accurate. Although they are time-consuming to
implement, the goal should be for the kinds of Earth-observation based adjustments described by Heinrich et al. 2023a for
Brazil to be achievable for all countries. This will ultimately facilitate comparisons between global models such as the GFW
model and NGHGIs, provide national policymakers with timely geospatial data in their own reporting terms, and build
confidence in the magnitude and trends of land-based anthropogenic emissions and sinks (Grassi et al. 2023).



## 4.3 Strengths and limitations of the GFW flux monitoring framework

The strengths of the current GFW flux model are broadly similar to those described in Harris et al. 2021. Strengths include its transparency, operational nature, flexibility, and updatability as new information becomes available. Here we focus on the complementarity of the GFW model with other land flux monitoring approaches. A strength of flux monitoring based on Earth observation, and therefore geospatial, data is its geographic specificity, while maintaining spatial consistency. Knowing where changes in land use and land cover—and the emissions and removals they have caused—occurred may help identify what factors are responsible for these changes and how to attribute them to specific human activities. While detailed information from ground surveys and activity data generated using local training data may provide more detail and accuracy at local scales, understanding the magnitude and distribution of global change requires a combination of both ground- and space-based observations (Houghton and Castanho 2023). In this sense, it fills in the gaps among other flux monitoring approaches. In terms of global consistency, the GFW model's key data are global in breadth and independent of data from the United Nations Food and Agriculture Organization, giving it a separate source for forest change data from bookkeeping models (Hansis et al. 2015, Gasser et al. 2020, Houghton and Castanho 2023). Moreover, by having an open-source model based on publicly available data, others can evaluate the model, make improvements, and/or adapt it to use national or local rather than global data. Users can keep some defaults while replacing others with better or more specific information, and understand how results are impacted by the various changes made for regions or at scales that interest them most.

Limitations are also broadly similar to those described in Harris et al. 2021. First, combining multiple spatially explicit data sources compounds the errors present in each individual source used in the framework. The GFW model partially manages this over larger areas through uncertainty propagation analysis to identify the relative contributions of different model components to uncertainty in each climate domain but cannot provide a pixel-level accuracy or uncertainty map. Extending the uncertainty framework to smaller regions (e.g., biomes or countries) would require uncertainty information for each of the individual data sources to be available at the desired scale of uncertainty propagation analysis. Second, the gain-loss approach of starting with baseline carbon densities and adding gains and subtracting losses over time has the potential to generate unrealistic estimates over longer periods due to drift from the original benchmark map. The GFW model could potentially address this through recalibration of carbon densities and forest extent at one or more intermediate years (e.g., 2010, 2015). Finally, the GFW model continues to have temporal limitations for both activity data and removal factors. The shorter gain period compared to tree cover loss in the original publication (12 vs. 19 years, respectively) has largely been addressed with the extension of tree cover gain through 2020, but the tree cover loss timeseries has its own inconsistencies (Weisse and Potapov 2021). The improvement in Earth observation data and changes to processing confounds apparent trends in gross emissions based on tree cover loss; it is difficult to determine how much the trends in emissions are due to real increases vs. better detection of disturbances through time. For removal factors, the concern is not so much temporal inconsistency as temporal constancy; the model makes the simplifying assumption of static removal factors, i.e. removal factors do not change





as forests grow or climate changes. Thus, the GFW model does not incorporate growth-response curves or climate feedbacks,
unlike in Earth System Models.

**4.4 Anticipated model developments**

Beyond annual updates to the GFW model, we anticipate continued, substantial changes to and research around both activity
data and emission and removal factors. These do not change the underlying conceptual framework but rather its implementation
as the model.
For activity data, anticipated model developments include:
1.  Global forest change data: The model will use annual forest extent, loss, and gain maps for greater temporal detail

(similar to Potapov et al. 2019 or Turubanova et al. 2023) and improved representation of carbon dynamics. For

example, the year of tree cover gain will be known (at least approximately) and repeated forest disturbances in the

same location will be captured (unlike in Hansen et al. 2013), allowing the generation of annual time series of gross

emissions, gross removals, and net flux. This should further enhance comparability of flux trends with the GCB and

NGHGIs.

2.  Drivers of forest loss: The model currently uses a global map of drivers of forest loss at 10-km resolution (Curtis et

al. 2018, updated to 2023) but research on mapping drivers of forest loss is advancing. An anticipated 1-km resolution

global map of drivers of forest loss will detect drivers that are not dominant at 10-km (and are therefore not mapped)

but are important at smaller scales, such as loss due to small-scale infrastructure and built-up areas amid loss due to

agricultural commodity expansion. Moreover, a separate driver class of forest loss due to natural disturbances would

further help with parsing natural and anthropogenic fluxes for translation into NGHGI reporting categories.

3.  Delineation of organic soils and their drainage status: The GFW model currently compiles several different data

sources (Table 2), which have different definitions and resolutions, to map organic soil extent. The GFW model would

benefit from a globally consistent organic soil map based on comprehensive aggregation of soil samples and

standardized mapping methods. However, it is not just the extent of organic soils but their drainage that affects

emissions in the GFW model. Thus, we are exploring improved mapping of organic soil drainage using recent

improvements in delineating road networks (OSM 2010; Meijer et al. 2018; Engert et al. 2024), drainage canal

networks (Dadap et al. 2021), and land cover (Potapov et al. 2022). More comprehensive maps of organic soil extent

and drainage will improve where the GFW model reports these emissions, particularly affecting non-$CO_2$ GHG

emissions.

4.  Improved initial forest age map: The GFW model currently classified forested pixels into primary forest, secondary

forest > 20 years, and secondary forest < 20 years old in 2000 using a few simple rules (described in Harris et al.



2021). However, a forest age map such as Besnard et al. 2021 could be used to refine the assignment of starting age
categories—particularly for secondary forests—or to determine where forest is along age-growth curves.
5.  Extent of planted forests and trees: The model currently uses SDPT v2.0 (Richter et al. 2024) but plans are underway
for SDPT 3.0, which will improve differentiation between natural and artificial stands in the United States and
Canada, along with other improvements for delineating planted tree extent in other countries.

For emission and removal factors, anticipated model developments include:
1.  Improved spatial and temporal resolution of forest carbon removals: The dominant role of removal factor uncertainties
in the uncertainty analysis highlights the need to further improve understanding of spatial and temporal variation in
forest carbon removals. Combining plot-level biomass estimates with spaceborne observations to produce static
biomass maps is well established (e.g., Saatchi et al. 2011, Santoro et al. 2021) and mapping biomass change is being
explored (Xu et al. 2021) but these do not provide spatiotemporally variable removal factors. An ecology-based, yet
still spatial, way to map removal factors could combine tree-level information collected in field plots with machine
learning methods to map forest population structure through time, including variables that influence biomass change
like upgrowth, mortality and recruitment for different forest types (Ma et al. 2020). Such an approach can generate
spatial and temporal predictions of how biomass changes across space and time that can be validated with forest plot
data. In conjunction with a time series of tree cover gain (in activity data list above), this would result in fully temporal
gross removals. Alternatively, growth curves for natural regeneration of forests could be developed, using methods
similar to Cook-Patton et al. 2020.
2.  Improved maps of soil carbon dynamics: The GFW model currently uses a benchmark map of soil organic carbon
density in mineral soil in 2000 and assumes loss of specific fractions of carbon under certain types of tree cover loss,
following a Tier 1 approach from IPCC 2019. However, a timeseries of soil organic carbon density in mineral soil
would support more realistic mapping of SOC dynamics.

Additionally, opportunities remain to compare GFW model emissions and removals with NGHGIs, bookkeeping models, and
regional or local data (e.g., Araza et al. 2023, Heinrich et al. 2023b). Such work would further our understanding of the
complementary roles of Earth observation-based forest carbon models and other approaches to forest flux monitoring.
**5 Data and code availability**
Gross emissions, gross removals, and net flux are available for download as 10x10 degree geotifs in 0.00025x0.00025-degree
resolution. Gross emissions files (Gibbs et al. 2024a) are at https://doi.org/10.7910/DVN/LNPSGP/ Gross removals files
(Gibbs et al. 2024b) are at https://doi.org/10.7910/DVN/V2ISRH. Net flux files (Gibbs et al. 2024c) are at
https://doi.org/10.7910/DVN/TVZVBI.    Data    are    also    available    as    assets    on    Google    Earth    Engine    at





https://code.earthengine.google.com/ae55707e335894d7be515386195390d2.      Code     is     available     at
https://github.com/wri/carbon-budget.

## 6 Conclusion

The updated Earth observation-based GFW forest carbon flux framework continues to show a substantial net sink for $CO_2$ in
forests globally, while also reporting sizeable gross emissions over half as large as gross removals since 2000. This highlights
ongoing opportunities to protect the forest carbon sink across a broad area and also reduce emissions from forest loss, especially
in hotspots of emissions that are discernable with our geospatial framework. The revised uncertainty analysis—with its
dramatic reduction in uncertainty in gross removals—demonstrates the importance of refining forest carbon sequestration rate
estimates. The flexibility of the model supports analyses at a range of spatial scales, while its operational nature means it can
incorporate new and existing Earth observation products and provide timely maps and data. Our translation of the GFW
model's fluxes into the reporting framework that NGHGIs use provides another lens through which to look at country-level,
land-based climate mitigation and is a resource for national policymakers interested in timely, spatial data on land fluxes. It
also demonstrates the two approaches' ability to improve, assess, and potentially confirm each other. Ultimately, confidence
and transparency are needed in assessments of progress towards the Paris Agreement, and Earth observation-based forest
carbon models are another tool to build consensus.

## Author contributions

NH and DAG conceived the model updates, and DG and MR executed the model updates. DAG executed the updated
uncertainty analysis. GG, SR, and JM created the national greenhouse gas inventory data set, and MR and GG compared it to
the GFW model. VH contributed to the translation of the GFW model for Brazil. DAG led manuscript preparation with
contributions from all co-authors.

## Financial support

This work received support from Norway's International Climate and Forest Initiative, the Bezos Earth Fund, and the Doris
Duke Foundation.

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

Methods, Version 1.1, 2024.

Grassi, G., House, J., Kurz, W. A., Cescatti, A., Houghton, R. A., Peters, G. P., Sanz, M. J., Viñas, R. A., Alkama, R., Arneth,
845       A., Bondeau, A., Dentener, F., Fader, M., Federici, S., Friedlingstein, P., Jain, A. K., Kato, E., Koven, C. D., Lee, D.,
Nabel, J. E. M. S., Nassikas, A. A., Perugini, L., Rossi, S., Sitch, S., Viovy, N., Wiltshire, A., and Zaehle, S.:
Reconciling global-model estimates and country reporting of anthropogenic forest CO2 sinks, Nature Clim Change,
8, 914–920, https://doi.org/10.1038/s41558-018-0283-x, 2018.

Grassi, G., Conchedda, G., Federici, S., Abad Viñas, R., Korosuo, A., Melo, J., Rossi, S., Sandker, M., Somogyi, Z., Vizzarri,
850       M., and Tubiello, F. N.: Carbon fluxes from land 2000–2020: bringing clarity to countries' reporting, Earth System
Science Data, 14, 4643–4666, https://doi.org/10.5194/essd-14-4643-2022, 2022.

Grassi, G., Schwingshackl, C., Gasser, T., Houghton, R. A., Sitch, S., Canadell, J. G., Cescatti, A., Ciais, P., Federici, S.,
Friedlingstein, P., Kurz, W. A., Sanz Sanchez, M. J., Abad Viñas, R., Alkama, R., Bultan, S., Ceccherini, G., Falk,
S., Kato, E., Kennedy, D., Knauer, J., Korosuo, A., Melo, J., McGrath, M. J., Nabel, J. E. M. S., Poulter, B.,
Romanovskaya, A. A., Rossi, S., Tian, H., Walker, A. P., Yuan, W., Yue, X., and Pongratz, J.: Harmonising the land-
use flux estimates of global models and national inventories for 2000–2020, Earth System Science Data, 15, 1093–
1114, https://doi.org/10.5194/essd-15-1093-2023, 2023.

Gumbricht, T., Roman-Cuesta, R. M., Verchot, L., Herold, M., Wittmann, F., Householder, E., Herold, N., and Murdiyarso,
D.: An expert system model for mapping tropical wetlands and peatlands reveals South America as the largest
contributor, Global Change Biology, 23, 3581–3599, https://doi.org/10.1111/gcb.13689, 2017.

Gunarso, P., Hartoyo, M., Agus, F. and Killeen, T.: Oil palm and land use change in Indonesia, Malaysia and Papua New
Guinea. Rep. Tech. Panels 2nd Greenh. Gas Work. Group Roundtable Sustain. Palm Oil RSPO 29-39, 2013.

Hansen, M. C., Potapov, P. V., Moore, R., Hancher, M., Turubanova, S. A., Tyukavina, A., Thau, D., Stehman, S. V., Goetz,
S. J., Loveland, T. R., Kommareddy, A., Egorov, A., Chini, L., Justice, C. O., and Townshend, J. R. G.: High-
Resolution Global Maps of 21st-Century Forest Cover Change, Science, 342, 850–853,
https://doi.org/10.1126/science.1244693, 2013.

Hansis, E., Davis, S. J., and Pongratz, J.: Relevance of methodological choices for accounting of land use change carbon fluxes,
Global Biogeochemical Cycles, 29, 1230–1246, https://doi.org/10.1002/2014GB004997, 2015.

Harris, N., Goldman, E. D., and Gibbes, S.: Spatial Database of Planted Trees (SDPT Version 1.0), 2019.
Harris, N. L., Gibbs, D. A., Baccini, A., Birdsey, R. A., de Bruin, S., Farina, M., Fatoyinbo, L., Hansen, M. C., Herold, M.,
Houghton, R. A., Potapov, P. V., Suarez, D. R., Roman-Cuesta, R. M., Saatchi, S. S., Slay, C. M., Turubanova, S. A.,
and Tyukavina, A.: Global maps of twenty-first century forest carbon fluxes, Nat. Clim. Chang., 11, 234–240,
https://doi.org/10.1038/s41558-020-00976-6, 2021.



Hastie, A., Honorio Coronado, E. N., Reyna, J., Mitchard, E. T. A., Åkesson, C. M., Baker, T. R., Cole, L. E. S., Oroche, C.
875         J. C., Dargie, G., Dávila, N., De Grandi, E. C., Del Águila, J., Del Castillo Torres, D., De La Cruz Paiva, R., Draper,
F. C., Flores, G., Grández, J., Hergoualc'h, K., Householder, J. E., Janovec, J. P., Lähteenoja, O., Reyna, D.,
Rodríguez-Veiga, P., Roucoux, K. H., Tobler, M., Wheeler, C. E., Williams, M., and Lawson, I. T.: Risks to carbon
storage from land-use change revealed by peat thickness maps of Peru, Nat. Geosci., 15, 369–374,
https://doi.org/10.1038/s41561-022-00923-4, 2022.
Heinrich, V., House, J., Gibbs, D. A., Harris, N., Herold, M., Grassi, G., Cantinho, R., Rosan, T. M., Zimbres, B., Shimbo, J.
Z., Melo, J., Hales, T., Sitch, S., and Aragão, L. E. O. C.: Mind the gap: reconciling tropical forest carbon flux
estimates from earth observation and national reporting requires transparency, Carbon Balance and Management, 18,
22, https://doi.org/10.1186/s13021-023-00240-2, 2023a.
Heinrich, V. H. A., Vancutsem, C., Dalagnol, R., Rosan, T. M., Fawcett, D., Silva-Junior, C. H. L., Cassol, H. L. G., Achard,
F., Jucker, T., Silva, C. A., House, J., Sitch, S., Hales, T. C., and Aragão, L. E. O. C.: The carbon sink of secondary
and degraded humid tropical forests, Nature, 615, 436–442, https://doi.org/10.1038/s41586-022-05679-w, 2023b.
Hengl, T., Jesus, J. M. de, Heuvelink, G. B. M., Gonzalez, M. R., Kilibarda, M., Blagotić, A., Shangguan, W., Wright, M. N.,
Geng, X., Bauer-Marschallinger, B., Guevara, M. A., Vargas, R., MacMillan, R. A., Batjes, N. H., Leenaars, J. G. B.,
Ribeiro, E., Wheeler, I., Mantel, S., and Kempen, B.: SoilGrids250m: Global gridded soil information based on
machine learning, PLOS ONE, 12, e0169748, https://doi.org/10.1371/journal.pone.0169748, 2017.
Holcomb, A., Burns, P., Keshav, S., and Coomes, D. A.: Repeat GEDI footprints measure the effects of tropical forest
disturbances, Remote Sensing of Environment, 308, 114174, https://doi.org/10.1016/j.rse.2024.114174, 2024.
Houghton, R. A. and Castanho, A.: Annual emissions of carbon from land use, land-use change, and forestry from 1850 to
2020, Earth System Science Data, 15, 2025–2054, https://doi.org/10.5194/essd-15-2025-2023, 2023.
Huang, Y., Ciais, P., Santoro, M., Makowski, D., Chave, J., Schepaschenko, D., Abramoff, R. Z., Goll, D. S., Yang, H., Chen,
Y., Wei, W., and Piao, S.: A global map of root biomass across the world's forests, Earth System Science Data, 13,
4263–4274, https://doi.org/10.5194/essd-13-4263-2021, 2021.
IPCC: IPCC Guidelines for National Greenhouse Gas Inventories, Prepared by the National Greenhouse Gas Inventories
Programme. Eggleston, H.S., Buendia, L., Miwa, K., Ngara, T. and Tanabe, K. (eds). IGES, Japan, 2006.
IPCC: 2013 Supplement to the 2006 IPCC Guidelines for National Greenhouse Gas Inventories: Wetlands. Hiraishi, T., Krug,
901         T., Tanabe, K., Srivastava, N., Baasansuren, J., Fukuda, M. and Troxler, T.G. (eds). IPCC, Switzerland, 2014a.
IPCC: Climate Change 2014: Synthesis Report. Contribution of Working Groups I, II and III to the Fifth Assessment Report
of the Intergovernmental Panel on Climate Change [Core Writing Team, Pachauri, R.K. and Meyer, L.A. (eds.)].
IPCC, Geneva, Switzerland, 151 pp, 2014b.



IPCC: 2019 Refinement to the 2006 IPCC Guidelines for National Greenhouse Gas Inventories, Calvo Buendia, E., Tanabe,
K., Kranjc, A., Baasansuren, J., Fukuda, M., Ngarize, S., Osako, A., Pyrozhenko, Y., Shermanau, P. and Federici, S.
(eds). IPCC, Switzerland, 2019.
IPCC: Climate Change 2022: Impacts, Adaptation, and Vulnerability. Contribution of Working Group II to the Sixth
Assessment Report of the Intergovernmental Panel on Climate Change [Pörtner, H.-O., Roberts, D.C., Tignor, M.,
Poloczanska, E.S., Mintenbeck, K., Alegría, A., Craig, M., Langsdorf, S., Löschke, S., Möller, V., Okem, A., and
Rama, B. (eds)]. Cambridge University Press. Cambridge University Press, Cambridge, UK and New York, NY,
USA, 3056 pp., doi:10.1017/9781009325844, 2022.
IPCC: 4th Corrigenda to the 2019 Refinement to the 2006 IPCC Guidelines for National Greenhouse Gas Inventories. Ed:
Sandro Federici, 2023
Ma, W., Lin, G., and Liang, J.: Estimating dynamics of central hardwood forests using random forests, Ecological Modelling,
419, 108947, https://doi.org/10.1016/j.ecolmodel.2020.108947, 2020.
MacCarthy, J., Tyukavina, A., Weisse, M. J., Harris, N., and Glen, E.: Extreme wildfires in Canada and their contribution to
global loss in tree cover and carbon emissions in 2023, Global Change Biology, 30, e17392,
https://doi.org/10.1111/gcb.17392, 2024.
Melo, J. et al. (in prep.) The LULUCF data hub: comparing land use emissions from national GHG inventories to other global
datasets.
Meijer, J. R., Huijbregts, M. A. J., Schotten, K. C. G. J., and Schipper, A. M.: Global patterns of current and future road
infrastructure, Environ. Res. Lett., 13, 064006, https://doi.org/10.1088/1748-9326/aabd42, 2018.
Miettinen, J., Shi, C., and Liew, S. C.: Land cover distribution in the peatlands of Peninsular Malaysia, Sumatra and Borneo
in 2015 with changes since 1990, Global Ecology and Conservation, 6, 67–78,
https://doi.org/10.1016/j.gecco.2016.02.004, 2016.
Mokany, K., Raison, R. J., and Prokushkin, A. S.: Critical analysis of root : shoot ratios in terrestrial biomes, Global Change
Biology, 12, 84–96, https://doi.org/10.1111/j.1365-2486.2005.001043.x, 2006.
Nabuurs, G-J., Mrabet, R., Abu Hatab, A., Bustamante, M., Clark, H., Havlík, P., House, J., Mbow, C., Ninan, K.N., Popp,
A., Roe, S., Sohngen, B., and Towprayoon, S.: Agriculture, Forestry and Other Land Uses (AFOLU). In Climate
Change 2022: Mitigation of Climate Change. Contribution of Working Group III to the Sixth Assessment Report of
the Intergovernmental Panel on Climate Change [Shukla, P.R., Skea, J., Slade, R., Al Khourdajie, A., van Diemen,
R., McCollum, D., Pathak, M., Some, S., Vyas, P., Fradera, R., Belkacemi, M., Hasija, A., Lisboa, G., Luz, S., and
Malley, J. (eds.)]. Cambridge University Press, Cambridge, UK and New York, NY, USA, 2022.
10.1017/9781009157926.009

Nabuurs, G.-J., Ciais, P., Grassi, G., Houghton, R. A., and Sohngen, B.: Reporting carbon fluxes from unmanaged forest,
Commun Earth Environ, 4, 1–4, https://doi.org/10.1038/s43247-023-01005-y, 2023.





Nyawira, S. S., Herold, M., Mulatu, K. A., Roman-Cuesta, R. M., Houghton, R. A., Grassi, G., Pongratz, J., Gasser, T., and
Verchot, L.: Pantropical CO2 emissions and removals for the AFOLU sector in the period 1990–2018, Mitig Adapt
Strateg Glob Change, 29, 13, https://doi.org/10.1007/s11027-023-10096-z, 2024.

Ochiai, O., Poulter, B., Seifert, F. M., Ward, S., Jarvis, I., Whitcraft, A., Sahajpal, R., Gilliams, S., Herold, M., Carter, S.,
Duncanson, L. I., Kay, H., Lucas, R., Wilson, S. N., Melo, J., Post, J., Briggs, S., Quegan, S., Dowell, M., Cescatti,
943            A., Crisp, D., Saatchi, S., Tadono, T., Steventon, M., and Rosenqvist, A.: Towards a roadmap for space-based
observations of the land sector for the UNFCCC global stocktake, iScience, 26, 106489,
https://doi.org/10.1016/j.isci.2023.106489, 2023.

Ogle, S. M., Domke, G., Kurz, W. A., Rocha, M. T., Huffman, T., Swan, A., Smith, J. E., Woodall, C., and Krug, T.:
Delineating managed land for reporting national greenhouse gas emissions and removals to the United Nations
framework convention on climate change, Carbon Balance and Management, 13, 9, https://doi.org/10.1186/s13021-
018-0095-3, 2018.

OSM (Open Street Map) roads and canals: Ramm, F., Topf, J., & Chilton, S. OpenStreetMap: Using and enhancing the free
map of the world. UIT Cambridge, 2010.

Pan, Y., Chen, J. M., Birdsey, R., McCullough, K., He, L., and Deng, F.: Age structure and disturbance legacy of North
American forests, Biogeosciences, 8, 715–732, https://doi.org/10.5194/bg-8-715-2011, 2011.

Pan, Y., Birdsey, R. A., Phillips, O. L., Houghton, R. A., Fang, J., Kauppi, P. E., Keith, H., Kurz, W. A., Ito, A., Lewis, S. L.,
Nabuurs, G.-J., Shvidenko, A., Hashimoto, S., Lerink, B., Schepaschenko, D., Castanho, A., and Murdiyarso, D.: The
enduring world forest carbon sink, Nature, 631, 563–569, https://doi.org/10.1038/s41586-024-07602-x, 2024.

Pearson, T. R. H., Brown, S., Murray, L., and Sidman, G.: Greenhouse gas emissions from tropical forest degradation: an
underestimated source, Carbon Balance and Management, 12, 3, https://doi.org/10.1186/s13021-017-0072-2, 2017.

Portugal. National Greenhouse Gas Inventory submitted to the UNFCCC, 1990-2018, 2020.
Potapov, P., Hansen, M. C., Laestadius, L., Turubanova, S., Yaroshenko, A., Thies, C., Smith, W., Zhuravleva, I., Komarova,
961            A., Minnemeyer, S., and Esipova, E.: The last frontiers of wilderness: Tracking loss of intact forest landscapes from
2000 to 2013, Science Advances, 3, e1600821, https://doi.org/10.1126/sciadv.1600821, 2017.

Potapov, P., Tyukavina, A., Turubanova, S., Talero, Y., Hernandez-Serna, A., Hansen, M. C., Saah, D., Tenneson, K.,
Poortinga, A., Aekakkararungroj, A., Chishtie, F., Towashiraporn, P., Bhandari, B., Aung, K. S., and Nguyen, Q. H.:
Annual continuous fields of woody vegetation structure in the Lower Mekong region from 2000-2017 Landsat time-
series, Remote Sensing of Environment, 232, 111278, https://doi.org/10.1016/j.rse.2019.111278, 2019.

Potapov, P., Turubanova, S., Hansen, M. C., Tyukavina, A., Zalles, V., Khan, A., Song, X.-P., Pickens, A., Shen, Q., and
Cortez, J.: Global maps of cropland extent and change show accelerated cropland expansion in the twenty-first
century, Nat Food, 3, 19–28, https://doi.org/10.1038/s43016-021-00429-z, 2022a.





Potapov, P., Hansen, M. C., Pickens, A., Hernandez-Serna, A., Tyukavina, A., Turubanova, S., Zalles, V., Li, X., Khan, A.,
Stolle, F., Harris, N., Song, X.-P., Baggett, A., Kommareddy, I., and Kommareddy, A.: The Global 2000-2020 Land
Cover and Land Use Change Dataset Derived From the Landsat Archive: First Results, Front. Remote Sens., 3,
https://doi.org/10.3389/frsen.2022.856903, 2022b.
Regulation - 2018/841 - EN - EUR-Lex: https://eur-lex.europa.eu/eli/reg/2018/841/oj, last access: 30 July 2024.
Richter, J., Goldman, E., Harris, N., Gibbs, D., Rose, M., Peyer, S., Richardson, S., and Velappan, H.: Spatial Database of
Planted Trees (SDPT Version 2.0), 2024.
Ruefenacht, B., Finco, M., Nelson, M., Czaplewski, R., Helmer, E., Blackard, J. A., Holden, G., Lister, A., Salajanu, D.,
Weyermann, D., and Winterberger, K.: Conterminous U.S. and Alaska Forest Type Mapping Using Forest Inventory
and Analysis Data, Photogrammetric Engineering & Remote Sensing, 74, https://doi.org/10.14358/PERS.74.11.1379,
2008.

Saatchi, S. S., Harris, N. L., Brown, S., Lefsky, M., Mitchard, E. T. A., Salas, W., Zutta, B. R., Buermann, W., Lewis, S. L.,
Hagen, S., Petrova, S., White, L., Silman, M., and Morel, A.: Benchmark map of forest carbon stocks in tropical
regions across three continents, Proceedings of the National Academy of Sciences, 108, 9899–9904,
https://doi.org/10.1073/pnas.1019576108, 2011.
Sanderman, J., Hengl, T., Fiske, G., Solvik, K., Adame, M. F., Benson, L., Bukoski, J. J., Carnell, P., Cifuentes-Jara, M.,
Donato, D., Duncan, C., Eid, E. M., Ermgassen, P. zu, Lewis, C. J. E., Macreadie, P. I., Glass, L., Gress, S., Jardine,
S. L., Jones, T. G., Nsombo, E. N., Rahman, M. M., Sanders, C. J., Spalding, M., and Landis, E.: A global map of
mangrove forest soil carbon at 30 m spatial resolution, Environ. Res. Lett., 13, 055002, https://doi.org/10.1088/1748-
9326/aabe1c, 2018.
Santoro, M., Cartus, O., Carvalhais, N., Rozendaal, D. M. A., Avitabile, V., Araza, A., de Bruin, S., Herold, M., Quegan, S.,
Rodríguez-Veiga, P., Balzter, H., Carreiras, J., Schepaschenko, D., Korets, M., Shimada, M., Itoh, T., Moreno
Martínez, Á., Cavlovic, J., Cazzolla Gatti, R., da Conceição Bispo, P., Dewnath, N., Labrière, N., Liang, J., Lindsell,
J., Mitchard, E. T. A., Morel, A., Pacheco Pascagaza, A. M., Ryan, C. M., Slik, F., Vaglio Laurin, G., Verbeeck, H.,
Wijaya, A., and Willcock, S.: The global forest above-ground biomass pool for 2010 estimated from high-resolution
satellite observations, Earth System Science Data, 13, 3927–3950, https://doi.org/10.5194/essd-13-3927-2021, 2021.
Schwingshackl, C., Obermeier, W. A., Bultan, S., Grassi, G., Canadell, J. G., Friedlingstein, P., Gasser, T., Houghton, R. A.,
Kurz, W. A., Sitch, S., and Pongratz, J.: Differences in land-based mitigation estimates reconciled by separating
natural and land-use $CO_2$ fluxes at the country level, One Earth, 5, 1367–1376,
https://doi.org/10.1016/j.oneear.2022.11.009, 2022.
Simard, M., Fatoyinbo, L., Smetanka, C., Rivera-Monroy, V. H., Castañeda-Moya, E., Thomas, N., and Van der Stocken, T.:
Mangrove canopy height globally related to precipitation, temperature and cyclone frequency, Nature Geosci, 12, 40–
45, https://doi.org/10.1038/s41561-018-0279-1, 2019.





Turubanova, S., Potapov, P., Hansen, M. C., Li, X., Tyukavina, A., Pickens, A. H., Hernandez-Serna, A., Arranz, A. P., Guerra-
Hernandez, J., Senf, C., Häme, T., Valbuena, R., Eklundh, L., Brovkina, O., Navrátilová, B., Novotný, J., Harris, N.,
and Stolle, F.: Tree canopy extent and height change in Europe, 2001–2021, quantified using Landsat data archive,
Remote Sensing of Environment, 298, 113797, https://doi.org/10.1016/j.rse.2023.113797, 2023.
Turubanova, S., Potapov, P., Hansen, M. C., Li, X., Tyukavina, A., Pickens, A. H., Hernandez-Serna, A., Arranz, A. P., Guerra-
Hernandez, J., Senf, C., Häme, T., Valbuena, R., Eklundh, L., Brovkina, O., Navrátilová, B., Novotný, J., Harris, N.,
and Stolle, F.: Tree canopy extent and height change in Europe, 2001–2021, quantified using Landsat data archive,
Remote Sensing of Environment, 298, 113797, https://doi.org/10.1016/j.rse.2023.113797, 2023.
Tyukavina, A., Potapov, P., Hansen, M. C., Pickens, A. H., Stehman, S. V., Turubanova, S., Parker, D., Zalles, V., Lima, A.,
Kommareddy, I., Song, X.-P., Wang, L., and Harris, N.: Global Trends of Forest Loss Due to Fire From 2001 to 2019,
Front. Remote Sens., 3, https://doi.org/10.3389/frsen.2022.825190, 2022.
UNEP-WCMC 2024, The World Database on Protected Areas (WDPA). Cambridge, UK: UNEP- WCMC, last accessed: 4
April 2024.
U.S. Department of Agriculture, Forest Service. The forest inventory and analysis database: database description and user
guide version 8.0 for Phase 2.
Vancutsem, C., Achard, F., Pekel, J.-F., Vieilledent, G., Carboni, S., Simonetti, D., Gallego, J., Aragão, L. E. O. C., and Nasi,
R.: Long-term (1990–2019) monitoring of forest cover changes in the humid tropics, Science Advances, 7, eabe1603,
https://doi.org/10.1126/sciadv.abe1603, 2021.
Weisse, M., Potapov, P.: How Tree Cover Loss Data Has Changed Over Time: https://www.globalforestwatch.org/blog/data-
and-tools/tree-cover-loss-satellite-data-trend-analysis, last access: 8 July 2024.
Xu, J., Morris, P. J., Liu, J., and Holden, J.: PEATMAP: Refining estimates of global peatland distribution based on a meta-
analysis, CATENA, 160, 134–140, https://doi.org/10.1016/j.catena.2017.09.010, 2018.
Xu, L., Saatchi, S. S., Yang, Y., Yu, Y., Pongratz, J., Bloom, A. A., Bowman, K., Worden, J., Liu, J., Yin, Y., Domke, G.,
McRoberts, R. E., Woodall, C., Nabuurs, G.-J., de-Miguel, S., Keller, M., Harris, N., Maxwell, S., and Schimel, D.:
Changes in global terrestrial live biomass over the 21st century, Science Advances, 7, eabe9829,
https://doi.org/10.1126/sciadv.abe9829, 2021.
Yang, F. and Zeng, Z.: Refined fine-scale mapping of tree cover using time series of Planet-NICFI and Sentinel-1 imagery for
Southeast Asia (2016–2021), Earth System Science Data, 15, 4011–4021, https://doi.org/10.5194/essd-15-4011-
1031      2023, 2023.

Zarin, D. J., Harris, N. L., Baccini, A., Aksenov, D., Hansen, M. C., Azevedo-Ramos, C., Azevedo, T., Margono, B. A.,
Alencar, A. C., Gabris, C., Allegretti, A., Potapov, P., Farina, M., Walker, W. S., Shevade, V. S., Loboda, T. V.,
Turubanova, S., and Tyukavina, A.: Can carbon emissions from tropical deforestation drop by 50% in 5 years?, Global
Change Biology, 22, 1336–1347, https://doi.org/10.1111/gcb.13153, 2016.



**Appendix A**
**Table A1. Comparison of forest carbon fluxes in Annex 1 countries, Non-Annex 1 countries, and globally between the GFW flux**
**model and national greenhouse gas inventories (NGHGIs).** Ranges in reported GFW values here come from two different scenarios: one
scenario where emissions from shifting agriculture in secondary forests is included in forest land, while the other scenario includes all
emissions from shifting agriculture in deforestation.

| | Net flux in forest land (Gt CO$_2$ yr$^{-1}$) | | Deforestation emissions (Gt CO$_2$e yr$^{-1}$) | | Net anthropogenic forest flux (Gt CO$_2$e yr$^{-1}$) | | Non-anthropogenic forest flux (Gt CO$_2$e yr$^{-1}$) | |
|---|---|---|---|---|---|---|---|---|
| | GFW | NGHGI | GFW | NGHGI | GFW | NGHGI | GFW | NGHGI |
| **Annex 1 countries** | -3.1 – -3.1 | -2.3 | 0.046 – 0.049 | 0.55 | -3.0 | -1.8 | -0.34 | N/A |
| **Non-Annex 1 countries** | -3.7 – -5.5 | -4.2 | 3.3 – 5.0 | 4.5 | -0.46 | 0.2 | -1.8 | N/A |
| **Global** | -6.8 – -8.5 | -6.6 | 3.3 – 5.0 | 5.0 | -3.5 | -1.6 | -2.2 | N/A |
