# Peer review of "Revised and updated geospatial monitoring of twenty-first century"

_Earth System Science Data, 2024_

## Referee Comment (RC1)

**Review: Gibbs et al. 2024 Revised and updated geospatial monitoring of twenty-first century forest carbon fluxes**

**General comments**

The authors present an update to the Global Forest Watch model that was first released in 2021. Model inputs have been improved, along with refined uncertainty analysis. In addition, the authors have gone to considerable effort to harmonise GFW estimates with NGHGI's, thus providing a means for national policymakers to assess their own reporting methods against a globally consistent, independent product. The paper is well written and structured, and I greatly appreciate the effort that has gone into making data and code not only accessible but usable and reproducible. After consideration of the specific comments below, I recommend publication in ESSD with minor revisions.

**Specific comments**

1. The abstract is quite long, and at times reads like a concluding paragraph. Consider revising.

2. How sensitive are the gross fluxes in the model to the forest cover definition? I noticed vast areas of more sparsely forested areas (but still forest by many country level definitions) in the drier regions of Africa and Australia are not currently mapped by GFW, and yet these regions are arguably more important for forest carbon uptake than the temperate regions simply owing to their enormous area. Is there a chance that by setting too high a threshold for canopy cover GFW is underestimating the impact of forest fluxes on the global carbon cycle? What might the implications of that be? Can you please clarify the reason for the 30 % canopy cover threshold?

3. The current GFW net flux map (to 2022) shows large areas of the forests in southeast Australia as either neutral or net emitters to the atmosphere presumably owing to the Black Summer Bushfires removing foliage cover, yet these areas have almost entirely (spectrally) recovered due to the high rainfall in the years after the fires (Rifai et al. 2024). I assume these forests are mapped as net emitters because the GFC product, as of the 2023 release, still labels these regions as 'deforested' (and the Potapov LULC dataset is static at 2020). In general, do you think the GFW model would underestimate carbon removals in forested ecosystems that are adapted to (somewhat) regular fire regimes? And on what timescale do you expect the inclusion of an annually updating forest cover gain product to be included in future model iterations (as mentioned in section 4.4)? Including some further discussion of this in the manuscript would be worthwhile to increase the users understanding of its limitations. For example, how does annual updating forest losses, but static forest gain, bias the net fluxes?

4. Section 2.3.2: Can you please clarify why forest bushfires in "Case 1" are considered anthropogenic emissions?

5. Table 4: Can the authors please consider including a column that compares fluxes between the original and updated versions of GFW over the same temporal period (2001-2019)? This would give the readers a quick sense of how much the change in fluxes is due to changes in the model inputs, versus fluxes accrued in the last few years (i.e. 2020-2022).

6. Figure 3a: Consider using a different colour palette as its hard to distinguish between low and high gross fluxes with the current pink-to-purple palette. Consider using instead one of the perceptually uniform sequential colourmaps.

7. Line 557:558. I'm not sure I agree that a comparison between GCB's 'all land' net terrestrial CO2 flux is a worthwhile comparison with GFW's net (high canopy cover) forest fluxes given the very different spatial extents those estimates represent. Is it possible instead to compare GFW's fluxes with a subset of the TRENDY DGVM fluxes masked to the same forest extent as GFW?

8. Line 742:744. Assuming a comparable model exists, why not include in this manuscript a comparison of GFW with a country/continental level estimate of net forest fluxes? That may help elucidate the strengths and limitations of the global model versus a regional model.

9. Is it possible to independently validate the model against a subset of eddy covariance flux towers in regions that haven't experienced disturbance? And could this comparison help quantify the differences between the gain-loss method (that may be limited in accounting for enhanced carbon uptake due to CO2 fertilisation), versus direct measurement of fluxes?

**Technical comments**

Line 747: DOI link is broken.

**References**

Rifai, Sami W., et al. "Burn severity and post-fire weather are key to predicting time-to-recover from Australian forest fires." Earth's Future 12.4 (2024): e2023EF003780.

---

## Author Comment (AC1)

**Reviewer 1:**

Review: Gibbs et al. 2024 Revised and updated geospatial monitoring of twenty-first century forest carbon fluxes

General comments

The authors present an update to the Global Forest Watch model that was first released in 2021. Model inputs have been improved, along with refined uncertainty analysis. In addition, the authors have gone to considerable effort to harmonise GFW estimates with NGHGI's, thus providing a means for national policymakers to assess their own reporting methods against a globally consistent, independent product. The paper is well written and structured, and I greatly appreciate the effort that has gone into making data and code not only accessible but usable and reproducible. After consideration of the specific comments below, I recommend publication in ESSD with minor revisions.

Specific comments

1) The abstract is quite long, and at times reads like a concluding paragraph. Consider revising.
   a) We have shortened the abstract and removed some of the concluding material.
2) How sensitive are the gross fluxes in the model to the forest cover definition? I noticed vast areas of more sparsely forested areas (but still forest by many country level definitions) in the drier regions of Africa and Australia are not currently mapped by GFW, and yet these regions are arguably more important for forest carbon uptake than the temperate regions simply owing to their enormous area. Is there a chance that by setting too high a threshold for canopy cover GFW is underestimating the impact of forest fluxes on the global carbon cycle? What might the implications of that be? Can you please clarify the reason for the 30 % canopy cover threshold?
   a) Our model does miss some emissions and removals by reporting fluxes for a canopy threshold of 30% instead of a lower value, like 10%. We have added global gross and net fluxes at canopy cover >10% to results section 3.1: "For example, defining forest as tree cover >10% instead of >30% (Hansen et al. 2013) results in gross emissions of 9.4 Gt CO2e yr-1, gross removals of -17.5 CO2 yr-1, and a net sink of -8.1 CO2e yr-1." We also added an explanation of the selection of the 30% threshold to the methods (section 2): "We use this definition of forests because a canopy density of >30% is a common threshold in for national definitions of forests (Harris et al. 2018) and because some of the input removal factors are applicable specifically to denser forest. All outputs and results use canopy density >30%, unless otherwise specified. However, because the model runs without any a priori canopy density threshold and

the forest definition is applied after the fact, fluxes can be estimated for lower canopy density thresholds."

3) The current GFW net flux map (to 2022) shows large areas of the forests in southeast Australia as either neutral or net emitters to the atmosphere presumably owing to the Black Summer Bushfires removing foliage cover, yet these areas have almost entirely (spectrally) recovered due to the high rainfall in the years after the fires (Rifai et al. 2024). I assume these forests are mapped as net emitters because the GFC product, as of the 2023 release, still labels these regions as 'deforested' (and the Potapov LULC dataset is static at 2020). In general, do you think the GFW model would underestimate carbon removals in forested ecosystems that are adapted to (somewhat) regular fire regimes? And on what timescale do you expect the inclusion of an annually updating forest cover gain product to be included in future model iterations (as mentioned in section 4.4)? Including some further discussion of this in the manuscript would be worthwhile to increase the users understanding of its limitations. For example, how does annual updating forest losses, but static forest gain, bias the net fluxes?

   a) This is an inherent limitation in the current model. We probably do underestimate post-disturbance removals, not just in fire-adapted regions but after any disturbance, because the tree cover gain map is one period (2000-2020) rather than temporally explicit. We have added a discussion of this in section 4.3 (strengths and limitations): "More limiting than the mismatch of tree cover loss and gain durations is the one-time nature of tree cover gain. Because the year of tree cover gain is not known, the model does not necessarily include post-disturbance gross regrowth and removals, which may underestimate removals and decrease the net sink. This effect would be particularly pronounced in forest where disturbance occurs earlier in the model and regrowth is substantial." Regarding the "timescale" when we expect the inclusion of annual gain data, we hope that it will be 2025.

4) Section 2.3.2: Can you please clarify why forest bushfires in "Case 1" are considered anthropogenic emissions?

   a) Case 1 countries explicitly or implicitly consider all land managed. According to IPCC guidance, "it is good practice to report all areas affected by disturbances such as fires, pest outbreaks and windstorms that occur in managed forest lands irrespective of whether these were the result of human activity" (IPCC 2003, https://www.ipcc.ch/site/assets/uploads/2018/03/GPG_LULUCF_FULLEN.pdf). Thus, fires anywhere in the country that occur on managed land constitute anthropogenic emissions in our flux reclassification system. Specific countries may use the UNFCCC natural disturbance provision in accounting for they Nationally Determined Contribution but they still report national emissions totals with and

without from natural disturbances. Accordingly, for this global analysis the division of anthropogenic and natural fluxes is based on managed vs. unmanaged land.

5) Table 4: Can the authors please consider including a column that compares fluxes between the original and updated versions of GFW over the same temporal period (2001-2019)? This would give the readers a quick sense of how much the change in fluxes is due to changes in the model inputs, versus fluxes accrued in the last few years (i.e. 2020-2022).

   a) We have added gross emissions from the revised model for 2001-2019. We cannot add gross removals and net flux for 2001-2019 using the updated model because those outputs are not available as a timeseries which can be subset to specific years. We have added a note on the table explaining this: "The revised model does not have gross removals and net flux values for 2001-2019 because they are an annual average over the entire model period rather than a timeseries and thus cannot be subset by year."

6) Figure 3a: Consider using a different colour palette as its hard to distinguish between low and high gross fluxes with the current pink-to-purple palette. Consider using instead one of the perceptually uniform sequential colourmaps.

   a) We have revised all three map panels (gross emissions, gross removals and net flux). We started with a divergent, color blind friendly color palette (https://colorbrewer2.org/#type=diverging&scheme=BrBG&n=10) for the net flux map (panel c), then applied the brown end to gross emissions map (panel a) and the teal end to gross removals map (panel b). This way, all three panels continue to be consistent and coherent in color and the gross fluxes use more sequential palettes than before.

7) Line 557:558. I'm not sure I agree that a comparison between GCB's 'all land' net terrestrial CO2 flux is a worthwhile comparison with GFW's net (high canopy cover) forest fluxes given the very different spatial extents those estimates represent. Is it possible instead to compare GFW's fluxes with a subset of the TRENDY DGVM fluxes masked to the same forest extent as GFW?

   a) We included the comparison with the GCB because we have been asked many times about how the GFW forest flux model relates to the GCB specifically and other global land flux estimates in general. We agree that there are important differences between the GCB and GFW forest flux model in terms of scope, which we describe in Section 4.1. Moreover, a comparison with DGVM fluxes masked to the GFW model extent would leave other key conceptual differences. In the GCB, DGVMs are used to estimate the land sink and don't produce estimates of land-use change emissions, so DGVMs on their own are also not conceptually comparable to our flux model.

8) Line 742:744. Assuming a comparable model exists, why not include in this manuscript a comparison of GFW with a country/continental level estimate of net forest fluxes? That may help elucidate the strengths and limitations of the global model versus a regional model.

   a) Such a comparison has been performed by Heinrich et al. 2023 for three countries (Brazil, Indonesia, Malaysia). Their detailed analysis elucidates strengths and limitations of our global model vs. country models. As we note in the discussion, country-level comparisons are beyond the scope of this update article because each country presents a unique situation and a thorough comparison that fully explains the differences can easily be a paper on its own. Splitting our results between Annex 1 and Non-Annex 1 countries was intended as a first sub-global disaggregation; many others are possible. Readers who want country-level comparisons can refer to the graphs of GCB, NGHGI, and GFW model fluxes on the JRC LULUCF Data Hub (added to discussion Section 4.2): "As an initial resource for country-level data, the European Union Joint Research Centre LULUCF Data Hub presents graphs of national land fluxes according to their NGHGIs, the Global Carbon Budget, and the translated fluxes from the GFW model (https://forest-observatory.ec.europa.eu/carbon/fluxes)."

9) Is it possible to independently validate the model against a subset of eddy covariance flux towers in regions that haven't experienced disturbance? And could this comparison help quantify the differences between the gain-loss method (that may be limited in accounting for enhanced carbon uptake due to CO2 fertilisation), versus direct measurement of fluxes?

   a) We appreciate the suggestion to compare our model results of net carbon gains or losses to those from flux towers, but it would not necessarily serve as a validation of our model, regardless of the results. We would need to identify flux towers that are only monitoring undisturbed forest fluxes (and do not include fluxes from other land uses or activities), identify the forest extent or footprint that contributes to the flux towers' measurements, and reconcile differences in temporal resolution/extent and processes included in each data source. For example, flux towers will not use the "committed approach" to emissions and thus be reflecting fluxes from decomposition over a longer "legacy" time period. Furthermore, the gain-loss method used in our global, geospatially explicit model is not designed to be an estimate that can contribute to a direct quantitative test against flux tower data at specific sites that measure net ecosystem exchange of $CO_2$ with the atmosphere. Even field-derived net primary productivity measurements collected at the same sites as flux towers, implemented correctly and comprehensively, are difficult to use for this purpose due to the precision needed (see e.g. Clark et al. 2001 for a

discussion on relating field-measured NPP data to whole-forest flux measurements). Such a comparison may be instructive in providing qualitative cross-checks and interpretations of both data sources but would be a separate research project beyond this data description paper.

Technical comments

Line 747: DOI link is broken.

      a. We have fixed the DOI link.

References

Rifai, Sami W., et al. "Burn severity and post-fire weather are key to predicting time-to-recover from Australian forest fires." Earth's Future 12.4 (2024): e2023EF003780.

**Reviewer 2:**

This manuscript presents updated global maps of greenhouse gas emissions and sequestration by forests using the Global Forest Watch (GFW) model from 2001. The authors also reconciled the GFW with national greenhouse gas inventories (NGHGIs).

Overall, the manuscript is well-written with clearly defined objectives, and the study itself is both interesting and well-executed.

There are a few minor points that could be addressed:

1) The abstract does not emphasize the key points effectively and could be more cohesive.

   a) We have revised the abstract.

2) What could be the potential reasons that cause the difference trends between the GFW and aggregated NGHGIs?

   a) We have added "The differing trends between the GFW flux model and aggregated NGHGIs is likely driven by generally increasing annual tree cover loss used in GFW (Hansen et al. 2013), as that has the greatest interannual variability present in either dataset." to Section 4.2.

3) Would using field measurements to validate the results help enhance their credibility?

   a) First, field measurements are included in the results insofar as they form the basis of the model's geospatial layers for aboveground biomass, belowground biomass, and soil carbon estimates. The benefit of the remote sensing-based approach used here is that many forests globally remain inaccessible to ground-based measurement and monitoring, and the time, costs and logistics involved in repeated measurements often prove prohibitive. Additional field measurements could support the credibility of the GFW forest flux model in specific places if plots can be found and made available with the same system boundaries, that track the same fluxes, and that operate at the same spatial and temporal scales. The most common field measurements that could be used to help enhance the credibility of model results and reduce uncertainties is additional plot measurements of annual increment of live aboveground biomass and soil carbon. In many cases, these data are not made publicly available or simply do not represent the same processes the GFW model does. We have an effort underway to compile additional forest plot data, which we cover in the discussion section under "anticipated model developments" (Section 4.4). Such a comparison with field measurements may be instructive but would be a separate research project beyond this data description

paper. As for comparison of our results with respect to flux tower data, see response to Reviewer 1, question #9.

**Citation**: https://doi.org/10.5194/essd-2024-397-RC2